# Comprehensive micro-scaled proteome and phosphoproteome characterization of archived retrospective cancer repositories

Corinna Friedrich [1,2,3,4,5], Simon Schallenberg[3], Marieluise Kirchner[6,7], Matthias Ziehm [6,7], Sylvia Niquet[6,7], Mohamed Haji[6], Christin Beier[6], Jens Neudecker[8], Frederick Klauschen [1,2,3,7,9 ✉] & Philipp Mertins [1,2,6,7 ✉]

Formalin-fixed paraffin-embedded (FFPE) tissues are a valuable resource for retrospective clinical studies. Here, we evaluate the feasibility of (phospho-)proteomics on FFPE lung tissue regarding protein extraction, quantification, pre-analytics, and sample size. After comparing protein extraction protocols, we use the best-performing protocol for the acquisition of deep (phospho-)proteomes from lung squamous cell and adenocarcinoma with >8,000 quantified proteins and >14,000 phosphosites with a tandem mass tag (TMT) approach. With a microscaled approach, we quantify 7,000 phosphosites, enabling the analysis of FFPE biopsies with limited tissue amounts. We also investigate the influence of pre-analytical variables including fixation time and heat-assisted de-crosslinking on protein extraction efficiency and proteome coverage. Our improved workflows provide quantitative information on protein abundance and phosphosite regulation for the most relevant oncogenes, tumor suppressors, and signaling pathways in lung cancer. Finally, we present general guidelines to which methods are best suited for different applications, highlighting TMT methods for comprehensive (phospho-)proteome profiling for focused clinical studies and label-free methods for large cohorts.

[1] German Cancer Consortium (DKTK), partner site Berlin, Berlin, Germany. [2] German Cancer Research Center (DKFZ), Heidelberg, Germany. [3] Institute of Pathology, Charité - Universitätsmedizin Berlin, corporate member of Freie Universität Berlin, Humboldt-Universität zu Berlin, and Berlin Institute of Health, Berlin, Germany. [4] Max Delbrück Center for Molecular Medicine in the Helmholtz Association (MDC), MDC graduate school, Berlin, Germany. [5] Humboldt Universität zu Berlin, Institute of Chemistry, Berlin, Germany. [6] Max-Delbrück-Center for Molecular Medicine in the Helmholtz Association (MDC), Proteomics Platform, Berlin, Germany. [7] Berlin Institute of Health at Charité – Universitätsmedizin Berlin, Berlin, Germany. [8] Department of Surgery - Campus Charité Mitte and Campus Virchow-Klinikum, Charité - Universitätsmedizin Berlin, Berlin, Germany. [9] Institute of Pathology, Ludwig-Maximilians-Universität München, Munich, Germany. ✉email: frederick.klauschen@charite.de; philipp.mertins@mdc-berlin.de

While genomics has transformed clinical cancer research and diagnostics, mutational profiles are often complex and require additional molecular methods such as proteomics for functional and clinical interpretation. Comprehensive proteomic profiling requires mass spectrometry, which is now broadly applied to study the consequences of genetic alterations in cancer on the proteome level and enable us to monitor changes in oncogenic signal transduction by global phosphoproteome profiling[1–4]. In an ideal scenario, it is the best practice for proteomic and, in particular, phosphoproteomic analysis to work with fresh frozen tumor tissues[5]. However, tumor collections of fresh frozen specimens are typically limited in overall sample numbers due to additional costs in sample collection and often lack comprehensive information on treatment and disease outcomes due to shorter follow-up times. Therefore, formalin-fixed, paraffin-embedded (FFPE) specimens from routine diagnostic laboratories are an attractive resource for retrospective studies because of their long-term stability and often available information on clinical outcomes. FFPE tissues are routinely collected in clinics for histological, immunohistochemical, and molecular diagnostics such as DNA and RNA sequencing. Resected tissues or needle biopsies are fixed with formaldehyde solution via the formation of crosslinks between proteins[6]. Following fixation, the samples are embedded in paraffin blocks for easy handling and long-term storage.

Efficient proteomics techniques have been developed and evaluated for the reversal of crosslinks and protein extraction from FFPE tissues[7–9]. Protein crosslinks are usually reversed by sample boiling, typically in the presence of primary amine-containing aqueous buffers, and proteins extracted in the presence of various detergents, such as sodium dodecyl sulfate (SDS)[8,10], sodium deoxycholate (SDC)[11], RapiGest[12] or organic solvents like trifluoroethanol (TFE)[13,14]. In all of these FFPE proteomics workflows, extracted proteins are later on digested with trypsin into peptides that are then analyzed by liquid chromatography-tandem mass spectrometry (LC-MS/MS). Many pre-analytical variables have been evaluated and no detrimental effect was found so far on protein abundance and quality due to storage times of FFPE specimens for ten or more years[13,15].

Nowadays, label-free proteomics methods in which peptides derived from FFPE samples are directly analyzed in a one sample per one LC-MS/MS run manner ("single-shot runs") can provide quantitative information for between 2000 and 5000 proteins[14,16,17] and are well suited for analysis of large clinical cohorts. An alternative to label-free methods are isotopic labeling techniques, such as tandem mass tag (TMT) reagents, that allow to chemically modify peptide samples with up to 16 different barcodes[18] and analyze these samples in a multiplexed manner per LC-MS/MS experiment. Due to the high level of multiplexing, more MS instrument time can be spent on a TMT-labeled multiplexed sample set, therefore frequently a second dimension of chromatography separation is used and TMT samples are pre-fractionated into several unique subfractions before LC-MS/MS analysis[19]. This approach has been used successfully in recent studies to profile FFPE tissues for ovarian cancer[10,13], where over 8,000 proteins have been quantified, and hepatocellular carcinoma[20], with over 5000 quantified proteins. The highest reported phosphoproteome coverage in FFPE samples so far was achieved with TMT-based methods, at a coverage of 8000 phosphopeptides derived from 3000 proteins[13]. Another benefit of TMT technology is its isobaric nature, which leads to a summed increase of peptide abundance across all multiplexed samples. This lead to the development of the booster channel approach, where one sample within the plex is loaded with 10–200 fold more peptide material compared to the other samples. TMT booster channel applications are attractive for

microscaled proteome analysis, as previously shown for primary cells cultured from pancreatic islet samples[21], and single-cell proteome analysis[22].

In this study, we focus on the comparison of the two major non-small cell lung cancer (NSCLC) subtypes adenocarcinoma (ADC) and squamous cell carcinoma (SCC) as a clinical use case to benchmark and further improve methods for high throughput, micro-scaled, and comprehensive proteome and phosphoproteome profiling. The presented techniques allow us to monitor most of the relevant oncogenes, tumor suppressors, and signaling pathways in lung cancer on the protein or phosphoprotein level, at starting amounts as little as provided by transbronchial needle biopsies. We also provide guidelines on what proteomics/phosphoproteomics methods to use for different sample sets and recommend the use of TMT approaches for comprehensive (phospho-)proteome profiling for focused clinical studies.

## Results

**Benchmarking protein extraction protocols for FFPE proteomics.** We compared three published protocols for protein extraction from FFPE samples, which differ in lysis buffer composition (different detergents) and subsequent sample cleanup, and performed proteome analyses from FFPE lung tissue (Fig. 1a). The first protocol published by Hughes et al.[10] (SDS-SP3) uses SDS, a strong detergent that interferes with LC-MS/MS and needs to be removed before analysis via paramagnetic beads SP3 (single-pot, solid phase-enhanced sample preparation) cleanup. The second protocol (SDC) relies on the weaker SDC detergent that can later be removed by centrifugation after acidification[11]. The direct trypsinization (DTR) protocol uses the commercially available RapiGest detergent which increases the enzymatic activity of trypsin and therefore does not need to be diluted or removed between samples lysis and digestion[12].

Four replicates of $1 \times 10\,\mu m$ FFPE slices (ca. 150 mm² tumor area on average) were processed for each of the three protocols and 1 µg peptide of each sample was injected for LC-MS/MS measurements. Over 3000 proteins were identified in single-shot analyses of the four replicates across all three protocols, with the majority of proteins identified in both the DTR and SDS-SP3 processed samples (Fig. 1b). Comparison of the coefficient of variation (CV) of the protein intensities over four replicates per protocol for the proteins identified across all three protocols shows that the SDS-SP3 protocol has by far the lowest mean CV (0.35) compared to DTR (0.77) and SDC (0.69) protocols (Fig. 1d). Another important validation parameter was the coverage of relevant proteins for NSCLC (Supplementary Table 2), e.g., immunohistochemical markers, like cytokeratins (KRT5/6/7, immunohistochemistry (IHC) stainings for ADC/SCC shown in Supplementary Fig. 1A and B), the differentiation-determinant transcription factor TTF1 (NKX2-1), or the surfactant protein B processing aspartic protease Napsin A (NAPSA), and known signaling proteins like the oncogenic kinase EGFR, and members of the antioxidant response KEAP1-NRF2 pathway (CUL3). Most of these relevant proteins were covered with both DTR and SDS-SP3 protocols, but additional proteins like the KEAP1 targeting E3-Ligase Cullin-3 (CUL3), the epithelial cell adhesion molecule (EPCAM), or the protein kinase AKT2 were exclusively found in SDS-SP3 data. Due to the best overall proteome coverage, lung cancer-relevant proteome coverage, and highest reproducibility across four replicates (Fig. 1b–d, Supplementary Data 1), the SDS-SP3 protocol is the best-performing protocol and was used for subsequent experiments.

As a first application, the SDS-SP3 protocol was used for label-free quantification (LFQ) comparison of 30 NSCLC cases (16 ADC

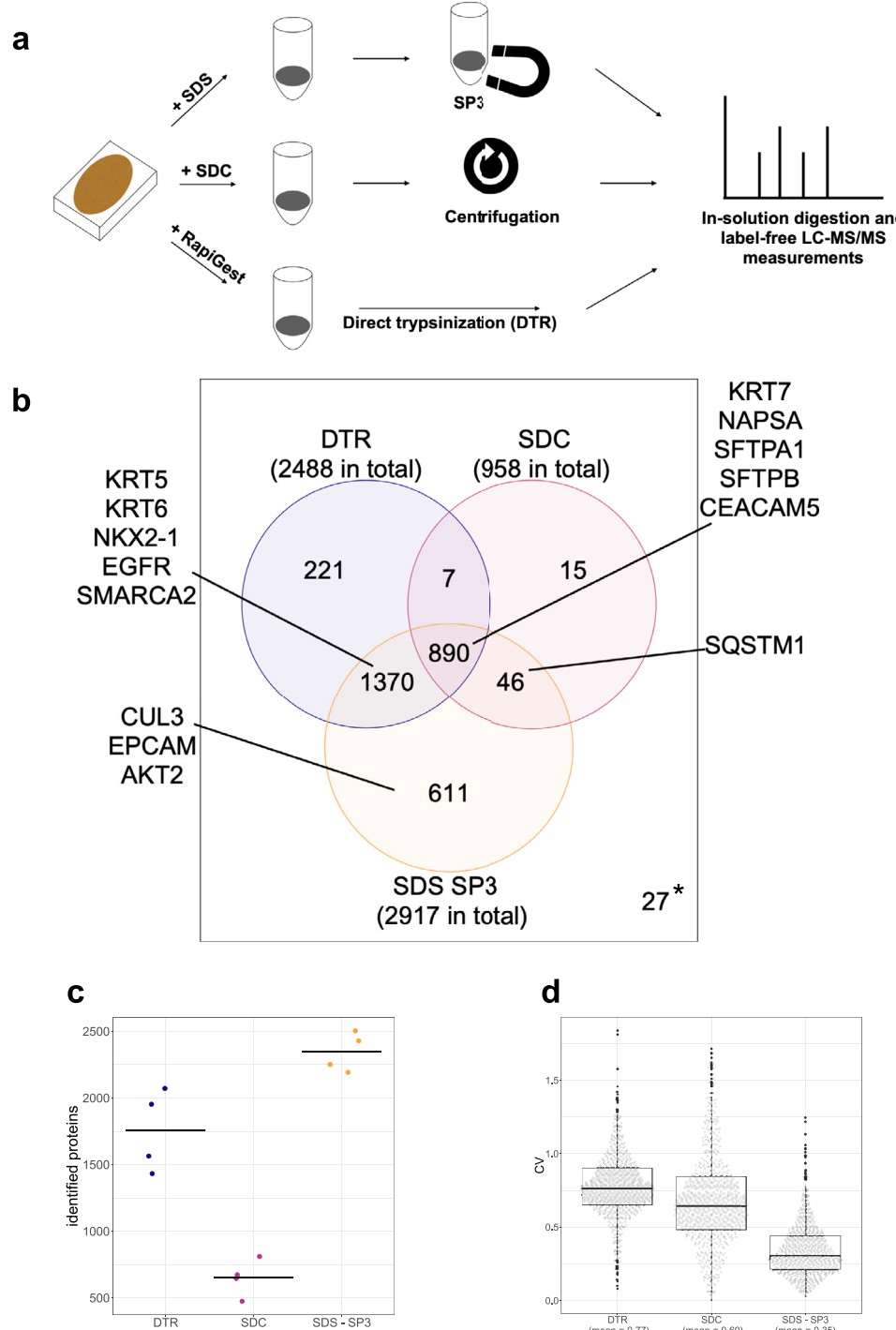

**Fig. 1 SDS-SP3 protocol performs best in terms of total identified proteins and identified NSCLC disease-related proteins. a** Comparison of three sample preparation protocols for FFPE proteome analysis. Deparaffinized FFPE lung tissues were lysed with different buffers containing either SDS, SDC, or RapiGest, and detergents were removed before digestion and LC-MS/MS analysis. **b** Overlap of proteins identified in DTR, SDC, and SDS-SP3 treated samples with NSCLC-relevant proteins highlighted as quality control. * marks proteins only identified but not quantified in any of the groups (27). **c** Proteins identified from a 1 μg single-shot injection from the samples processed with the three protein extraction protocols, the black bar is showing the mean of the group. Source data are provided as a Source Data file. **d** Boxplots plots showing the distribution of coefficients of variation (CV) per protein across four replicates for each protocol with boxplots. The plot depicts the 25th and 75th quartile (box), the median (thick black line), the minimum and maximum (whiskers, Q1-1.5*IQR, Q3 + 1.5*IQR), and outliers (dots). Source data are provided as a Source Data file.

and 14 SCC, Supplementary Table 1, Supplementary Data 2). Over 300 proteins out of 1728 quantified proteins were found to be significantly different between the two groups with ADC markers like KRT7 and SCC markers like KRT5/6 being significantly upregulated in their respective entity (Supplementary Fig. 2A). With this dataset, we could confirm several markers on a protein level that had previously been shown on RNA level[23], such as mucin-5B (MUC5B), a glycoprotein that is secreted in the lung and

is associated with poor prognosis in ADC[24], kininogen 1 (KNG1), and serum transthyretin (TTR), which has been shown to be associated with poor outcome[25]. In SCC we could confirm apolipoprotein 1 (APOA1) which has previously been described as inversely correlated with the risk of lung cancer[26]. Interestingly, the paper by Venugopal et al.[23] showed fibrinogen alpha (FGA) to be overexpressed in ADC, in our dataset, however, it was found with higher expression levels in SCC. In addition, we were able to quantify several proteins that might be of interest as potential future markers for lung ADC, such as mucin-5AC (MUC5AC), another mucin that has been linked to ALK-positive lung ADC[27], or carcinoembryonic antigen-related cell adhesion molecule 6 (CEA-CAM6), a glycoprotein that is involved in cell invasion and metastasis and has been shown to have higher expression levels in ADC compared to SCC via IHC[28].

Cystatin A (CSTA) has previously been shown to have a higher expression in SCC via IHC[29] and might be an interesting potential candidate for SCC.

A subset of these 30 samples consisting of 5 ADC and 5 SCC cases with sufficient sample amounts for subsequent TMT proteomic and phosphoproteomic method comparisons was selected. When ADC vs. SCC label-free proteome differences were compared in this subset of 10 samples, over 30 proteins were significantly upregulated in ADC vs. SCC, among which NSCLC-relevant proteins such as NAPSA and NQO1, another member of the Keap1-Nrf2 pathway, were represented (Supplementary Fig. 2B).

The "match between runs" (MBR) feature of the MaxQuant platform[30] was used to maximize the number of quantified proteins in the LFQ experiment. To ensure that the majority of quantified proteins is still identified from MS/MS spectra, we compared the log2-transformed LFQ intensity distributions of all quantified proteins to the subset of proteins with MS/MS spectra-based identifications and those only identified with the MBR feature (Supplementary Fig. 3A). The vast majority of quantified proteins was identified via MS/MS spectra (90%) and in this comparison, the MBR algorithm contributes only a small part of lower-intensity proteins (10%). This can also be seen in the individual ADC (Supplementary Fig. 3B) and SCC (Supplementary Fig. 3C) technical replicates where proteins identified only by spectral matching tend to have lower LFQ intensities and higher variability. Scatter plots show two technical replicates for all quantified proteins, for significantly differentially expressed proteins, and also for NSCLC markers used as quality controls in this study (Supplementary Fig. 3B/3C left to right). The average technical correlation for replicate pairs was 0.995 and 0.994 ($p < 0.01$) for ADC and SCC, respectively. The average biological correlation for ADC or SCC tumor within-group comparisons were 0.888 and 0.917, respectively.

Proteins that fall outside a 95% prediction interval in the scatterplot of technical ADC or SCC replicates are indicated in the volcano plots (Supplementary Fig. 2A/2B) with an asterisk symbol (301 out of 5059 proteins) whereby the variability is usually higher in proteins with lower abundance. We observe that none of the NSCLC-relevant proteins fall outside of these intervals.

**Assessment of pre-analytical variables for FFPE proteome profiling.** Tissue samples in the routine pathology labs can have different lengths of fixation time, ranging from overnight fixation (~24 h) during the week to over-the-weekend fixation (72 h). Therefore it is important to investigate the influence of different fixation times on sample quality. To test this, our cohort of 30 NSCLC cases was used. 18 (10 ADC, 8 SCC) samples from the cohort had been fixed in formalin for 24 h and 12 (6 ADC, 6 SCC)

samples had been fixed for 72 h. No significant differences in the number of identified proteins were observed between 24 and 72 h fixation time (Supplementary Fig. 1C). A moderated $t$-test comparing the protein quantities in the samples that were fixed overnight to those fixed from Friday to Monday showed only two proteins significantly upregulated (adjusted $p$-value < 0.05) in the 24 h samples. One was Clusterin (CLU) and the other one immunoglobulin heavy constant mu (IGHM), both variable extracellular proteins that are difficult to quantify (Supplementary Fig. 1D). This shows that there is no relevant bias in protein groups detected after 24 or 72 h formalin fixation.

In addition, we investigated the impact of fixation time on the number of formalin-induced modifications. Formalin-fixation has been reported to cause several protein modifications such as dimethylation, formylation, or the addition of a methylol group[31], and the storage of samples at room temperature may also cause methionine oxidation. A comparison of the number of identified peptides with each modification in samples that had been fixed either 24 or 72 h showed no significant time-dependent difference for any modification tested (Supplementary Fig. 1E). Based on these results, we can conclude that for the study design for proteomic analyses of larger clinical FFPE cohorts, fixation time does not have to be a selection criterion.

Next, we investigated the impact of heat on the efficiency of the de-crosslinking and the stability of the proteome and phospho-proteome. To avoid sample processing conditions that would lead to protein degradation or loss of phosphorylation, we compared different heat incubation times in a cell culture experiment. HEK293 cells were either fresh-frozen directly after harvesting, formalin-fixed immediately on the plate, and then harvested or first incubated at 4 °C for an hour and then formalin-fixed to mimic processing delays in the routine pathological lab. All samples were treated with the same SDS-lysis buffer and replicates of each type of sample were incubated either 10 min, 30 min, 60 min, or 120 min at 95 °C (Supplementary Fig. 4A), cleaned up with SP3, digested, and run in single-shot analyses. On peptide level, the two sample groups treated with formalin (cells fixed immediately and cells fixed after 1 h) showed an increase in identifications of up to 20% with increased heat incubation time, while for fresh-frozen cells the number of identifications decreased by 20% when incubated 120 min instead of 10 min. In general, only slightly fewer peptides were identified from the fixed cells at all time points (Supplementary Fig. 4B).

Equal amounts of each sample (100 μg) were enriched for phosphopeptides by immobilized metal affinity chromatography (IMAC) and in all three types of samples, the lowest number of phosphopeptides was identified after 10 min of cooking at 95 °C. Identification rates increased up to 50% with the maximum number of phosphopeptides identified at 60 min, then the rates decreased again slightly (Supplementary Fig. 4C). Noteworthy, phosphorylation modifications on proteins are stable enough to withstand heat incubation times of up to 2 h at 95 °C without any major abundance losses.

These results suggest that a compromise has to be found between the maximum number of identified peptides or phosphopeptides and the workflow has to be tailored to the individual application. For this study, we chose to keep the heat incubation step at two hours to maximize protein yield.

In addition, a TMT comparison of five replicates of fresh-frozen HEK293 cells and five replicates of formalin-fixed HEK293 cells showed a high correlation between the proteomes of fresh and formalin-fixed cells (Supplementary Fig. 5A) with Pearson correlation coefficients of 0.96–0.99 ($p < 0.01$). The phosphoproteomes of fresh-frozen and formalin-fixed cells also showed a good correlation with Pearson correlation coefficients of 0.7–0.9 ($p < 0.01$), an expected trend, considering that the

variability is generally higher on the (phospho-)peptide level (Supplementary Fig. 5B).

**Increased proteome/phosphoproteome coverage with multiplexed TMT.** TMT multiplexing, combined with prefractionation of peptide samples prior to LC-MS/MS allows to massively increase the coverage of proteomics experiments while keeping the instrument time used for individual tumor samples similar to classical single-shot label-free experiments. Usually, equal peptide amounts, here 200 µg per sample, are used for isotope labeling with isobaric compounds, combined and fractionated into multiple fractions over a high pH gradient. This separation method is orthogonal to the reversed-phase separation at acidic pH in the downstream nanoflow LC system and helps reduce overall sample complexity in each resulting sample fraction. Similar to previously developed workflows[19], we used 10% of the total peptide material for the proteome analysis and reserved 90% of the material for IMAC phosphopeptide enrichment and phosphoproteome analysis. The resulting fractions were analyzed by LC-MS/MS and individual samples were quantified using reporter ions derived from their isotope tags (Fig. 2a).

With this equal loading TMT approach, we were able to quantify on average 9000 proteins per TMT-plex experiment and an intersection of 8577 proteins corresponding to 8200 genes without missing values across two TMT11-plexes from FFPE lung tissue (Supplementary Data 3). The average measuring time per individual sample was 6 h. This is an increase of over 6000 proteins compared to the LFQ approach for the same set of samples. While many proteins relevant to NSCLC (e.g., KRT5/6/7, NAPSA, EPCAM, EGFR) were identified with both the label-free and equal loading TMT approach, many additional markers such as tumor protein p63, another SCC marker (TP63), NKX2-1, NRF2, KEAP1, and CDKN1A (p21), the cyclin-dependent kinase (CDK) inhibitor, which is also involved in both the KEAP1-NRF2 pathway and PI3K-AKT pathway[32,33] were only identified in the TMT dataset (Fig. 2b). The proteins identified in the equal loading TMT dataset cover almost 10,000 proteins over six orders of magnitude of summed peptide precursor intensities whereas the LFQ coverage limit is at about 1800 proteins, leaving many cancer-related proteins or oncogenes like the tumor suppressor p53 (TP53), oncogenic kinase BRAF and KEAP1 uncovered (Fig. 2c). Concerning the PI3K-AKT signaling pathway, which plays an important role in NSCLC, because it is involved in EGFR receptor tyrosine kinase inhibitor (RTKI) resistance[34], most members of the pathway were quantified in the equal loading TMT dataset (Fig. 3a).

For the phosphoproteome, over 14,000 phosphosites were quantified in the equal loading TMT dataset, corresponding to 4400 genes and four orders of magnitude in reporter ion intensities (Fig. 3b, Supplementary Data 4). The deep phosphoproteome adds additional information, as some proteins such as the oncogene Myc were only quantified in the phosphoproteome (Fig. 3a). In addition to quantifying proteins in their phosphorylated proteoform that have not been found in the proteome data, there is additional information on pathway regulation in the phosphoproteome data, for example, in cases where phosphosites show abundance differences where the corresponding proteins do not. A comparison of the log2 SCC/ADC fold changes of the proteome and phosphoproteome indicated that several proteins are differentially phosphorylated between ADC and SCC while the protein expression remains unchanged (Supplementary Fig. 6A). We then performed a gene ontology molecular function (GO MF) analysis on these proteins, which identified two groups, 1) proteins involved in transcription including the transcription factor FOXO3, or 2)

proteins involved in signal transduction, e.g., SOS1 (Supplementary Fig. 6B). SOS1 has been shown to play a role in tumor growth in lung ADC[35], while FOXO3 acts as a tumor suppressor by regulating apoptosis[36]. These results show that the standard TMT technology can be successfully applied to clinical FFPE samples, providing relevant information on protein expression and phosphorylation levels.

**Microscaled analysis for biopsy equivalents with a booster TMT sample.** The equal loading TMT approach is well suited for studies in resected FFPE tissues where the sample amount is not a concern, such as when multiple 10 µm slices of a $5 \times 5$ mm² tumor section are available. To investigate the applicability of the TMT approach for low sample amounts, such as needle biopsy FFPE samples, we utilized 20 µg instead of 200 µg aliquots of the same peptide samples used for the equal loading TMT experiment as "biopsy equivalents". Protein yield can vary between samples and 20 µg is an amount that one should be able to reliably extract from a large group of clinical biopsy samples. Eight of the biopsy equivalents were labeled with TMT (channels 1–8) and combined with a boosting channel of 2 mg internal standard (channel 11). To avoid carryover due to isotopic impurities of the labels, two TMT channels between the samples and 100-fold more abundant internal standard (channel 9 and 10) were left empty. The precursor intensity of each peptide in the MS1 scans is here a sum of the intensities across all samples/TMT channels (Fig. 4a). The boosting channel helps increase the signal over the minimal abundance threshold that is necessary to obtain enough fragment ions to determine the amino acid sequence of TMT-labeled peptides and enough TMT reporter ions for quantification in MS2 scans.

We were able to quantify on average 7712 proteins per TMT plex and over 7000 proteins without missing values with this deep proteome profiling approach for microscaled TMT samples (Supplementary Fig. 7A, Supplementary Data 5), covering five orders of magnitude in precursor intensities and biologically important proteins such as NAPSA, KRAS or KEAP1. The log2 SCC vs. ADC fold changes of equal loading and microscaled TMT correlated well with a Pearson correlation coefficient of 0.67 and a $p$-value below $2.2 \times 10^{-16}$, which shows that the boosting channel does not skew intensity ratios in the microscaled experiment (Fig. 4b). To better understand the quantified protein networks on a biological level, we performed a single-sample gene set enrichment analysis (ssGSEA[37]) with the "hallmarks of cancer" gene sets[38]. The hallmark pathway "protein secretion" was significantly upregulated in ADC samples (Supplementary Fig. 7B). This is consistent with what is known about lung cancer biology as lung ADC is believed to develop from club cells and alveolar epithelial type 2 cells, secretory cells in the bronchial epithelium[39]. These cells secrete a mix of proteins and other molecules to protect the epithelium[40,41]. The hallmark "inflammatory response" and several immune system-related terms are upregulated in SCC samples (Supplementary Fig. 7B). SCC is generally more associated with smoking. Cigarette smoke contains free radicals that induce oxidative stress and cell damage, which in turn cause inflammation[42].

While 14,000 phosphosites were quantified among the two equal loading TMT plexes, 8,800 phosphosites were still quantified among the two microscaled TMT plexes (Supplementary Data 6). An overlap of 6000 phosphosites was quantified in both the TMT equal loading and microscaled experiments. Among these, phosphorylation events on oncogenes like AKT1, MET and, mTOR were found. Almost 3000 phosphosites were quantified in the microscaled TMT experiment only, e.g., additional phosphosites on MET or TP63 (Fig. 4c).

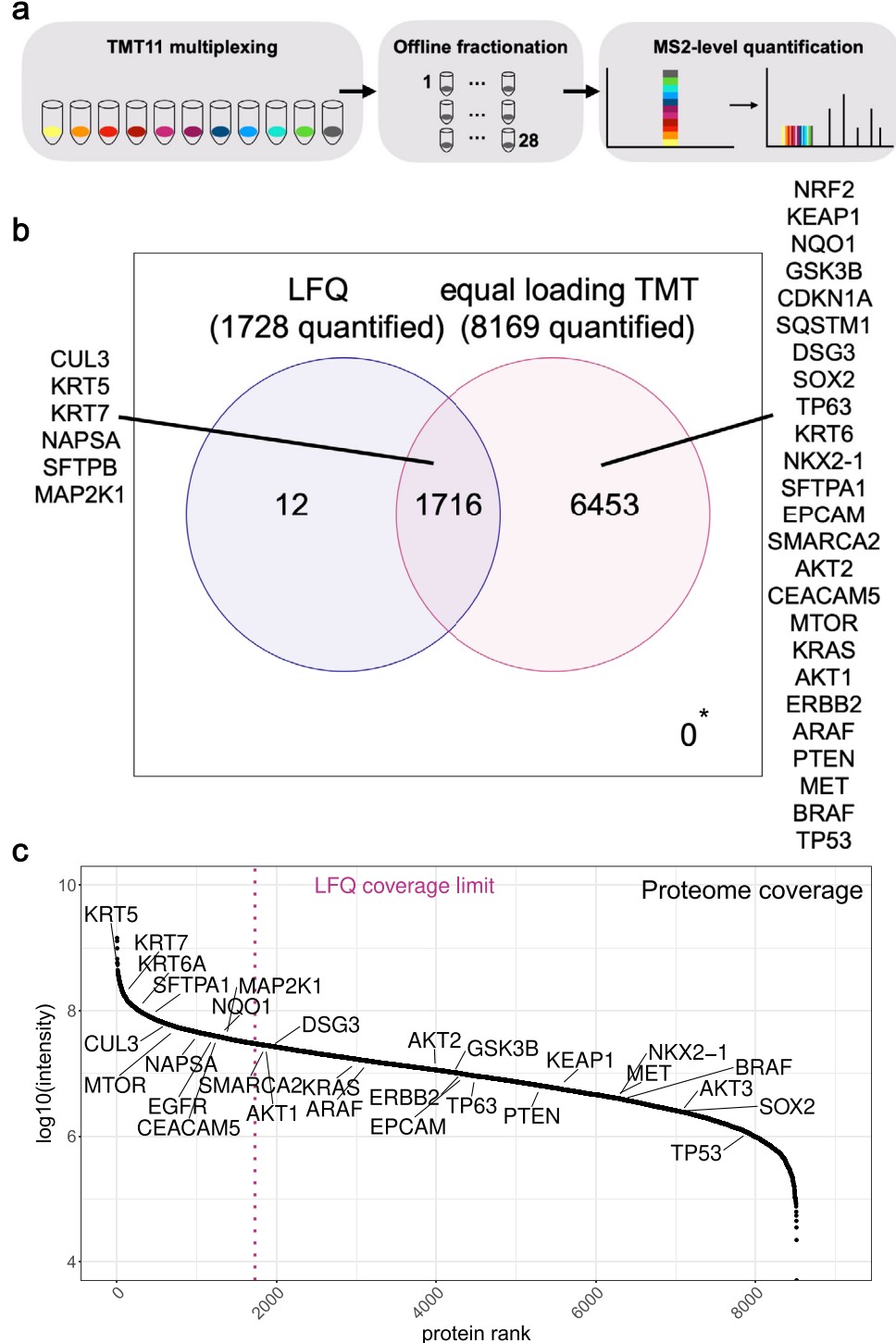

**Fig. 2 Deep FFPE proteome coverage with TMT11 labeling and two-dimensional liquid chromatography. a** TMT multiplexing combined with high pH offline fractionation allows for deeper LC-MS/MS coverage while spending less measuring time per individual sample. Multiplexed, TMT-labeled peptide samples are loaded in equal total quantities per TMT-channel and fractionated via two dimensions of liquid chromatography before MS analysis. Increased MS time requirements per TMT experiment for 28 injections are compensated here by multiplexed analysis of up to 11 samples. **b** Venn diagram showing the overlap of quantified proteins from LFQ and equal loading TMT quantification. NSCLC-relevant markers are shown as quality control. Only unique proteins and no isoforms were counted as quantified proteins. * marks proteins only identified but not quantified with either method (0). **c** Log10 reporter ion intensity distribution over all proteins that were quantified in equal loading TMT. The LFQ coverage limit is shown in magenta.

The observed coverage differences between equal loading and microscaled TMT phosphoproteomes can be attributed to multiple factors. First, the increased complexity of internal standard containing microscaled TMT samples coincides with decreased identifications on protein and phosphosite levels by 9.4% and 16%, respectively (Supplementary Fig. 8A/B). Due to higher sample amount requirements for the internal standard, a mix of 20 different NSCLC tumors was generated, which is more complex than the individual 10 tumors in the equal loading experiment. Second, in the microscaled TMT experiment, we

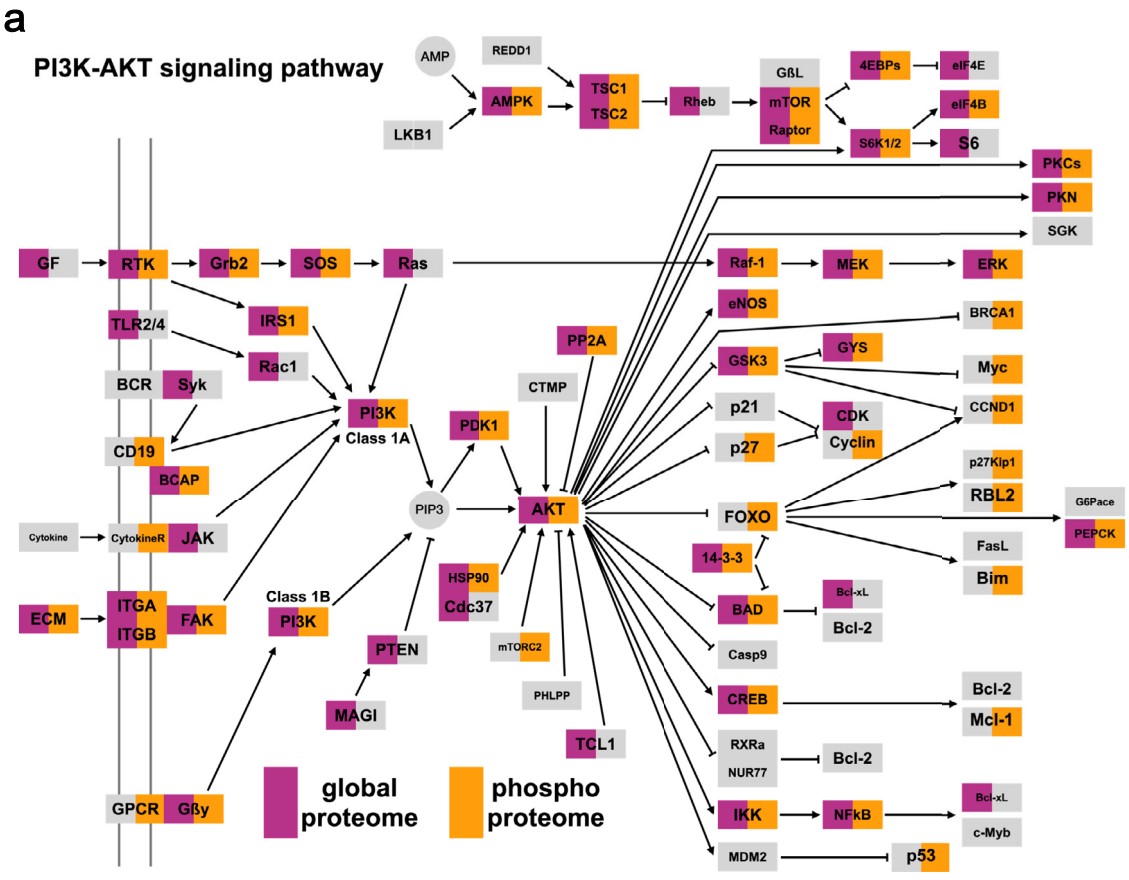

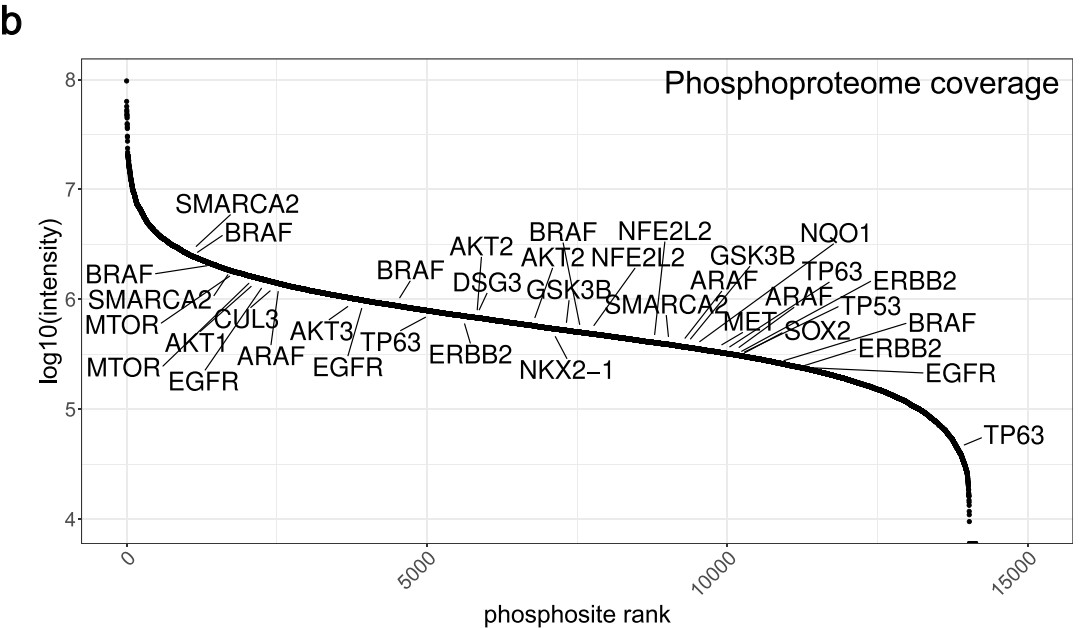

**Fig. 3 Phosphoproteome analysis with equal loading TMT11 provides a comprehensive overview of cancer-relevant pathways. a** Most key components of the PI3K-AKT signaling pathway were covered by equal loading TMT on a global proteome (shown in magenta) and phosphoproteome (shown in orange) level. PI3K-Akt signaling pathway was adapted from KEGG pathway hsa04151. **b** Log10 reporter ion intensity distribution over all phosphosites that were quantified in equal loading TMT. Lung cancer-relevant phosphoproteins are indicated by gene names. A coverage of 15,015 and 15,486 phosphosites was achieved for the two replicates (overlap of 14,133 phosphosites).

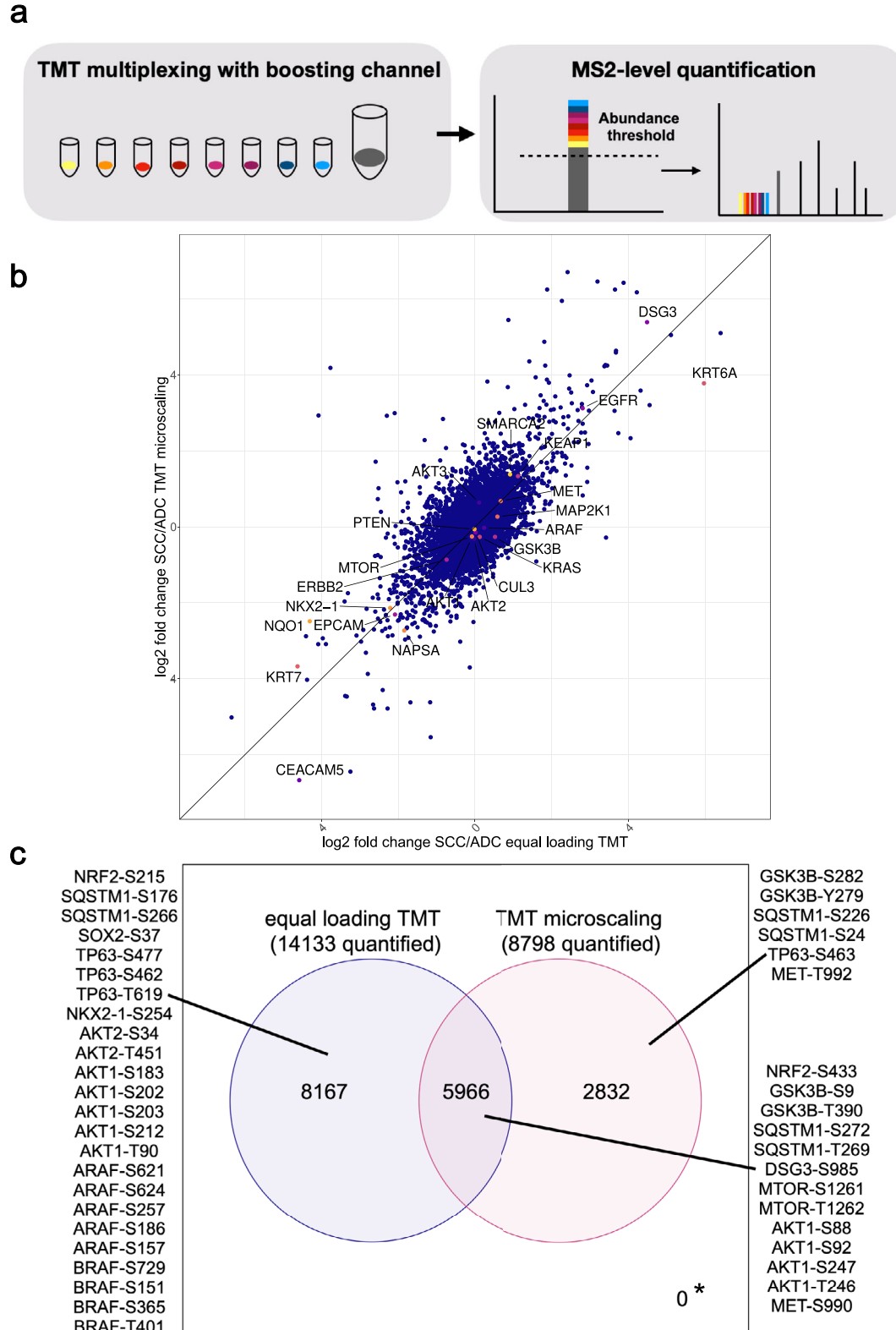

**Fig. 4 Deep proteome coverage even from small sample amounts by using a TMT booster sample. a** TMT11 combined with a boosting channel allows for microscaled deep proteome and phosphoproteome analysis of FFPE samples with low sample amounts (20 µg per sample). **b** Correlation of log2 fold changes of equal loading TMT and micoscaled TMT experiments in FFPE lung cancer samples shows similar quantification results (Pearson correlation coefficient $r = 0.67$, $p < 0.01$). NSCLC-relevant proteins are highlighted, all other proteins are shown in blue. Source data are provided as a Source Data file. **c** Venn diagram showing the overlap in quantified phosphosites between equal loading TMT and microscaled TMT with highlighted phosphosites on NSCLC-relevant proteins and oncogenes. * marks phosphosites not quantified with either method (0).

observed a 10.9% decrease going from identified to quantified phosphosites, whereas there is only a 0.75% difference in the equal loading TMT. These dropouts are caused by loss of signal in the tumor TMT channels, while a signal is still present in the internal standard TMT channel (100× more input). Third, comparing the two replicate plexes for each TMT experiment (px1 and px2 for equal loading TMT and px3 and px4 for microscaled TMT), the percentage of phosphosites reproducibly quantified in both plexes is higher in the equal loading TMT than in the microscaled TMT (85.7% vs 63.5%, Supplementary Fig. 8A/8B). The 22.2 percentage points difference is likely due to the higher variability of lower abundant phosphopeptides in microscaled TMT.

The hydrophobicity index[43] for the phosphopeptides quantified in equal loading and microscaled TMT is distributed very similarly, both for the total phosphoproteome (Supplementary Fig. 8C) and the subset of phosphopeptides that were either uniquely quantified in the equal loading TMT dataset or the microscaled TMT dataset (Supplementary Fig. 8D, see also Venn diagram Fig. 4c). This suggests that the two TMT approaches do not show a bias with respect to the type of phosphopeptides that are enriched and subsequently quantified, but that the differences arise merely from the variability that is known from the stochastic detection of peptides in data-dependent acquisition analysis of low abundant phosphopeptide samples.

With the microscaled TMT approach with only 20 µg of peptides from each channel and a boosting channel with 100× more input material, we could still cover most of the PI3K-Akt pathway (Supplementary Fig. 9A). Here, more proteins were only covered in their phosphorylated form than in the equal loading dataset. Examples are p27 (cyclin-dependent kinase inhibitor 1B, CDKN1B), FOXO3, mTORC2, and p53. This shows that phosphopeptide enrichment offers additional information, mostly about proteins with low abundance, as was shown by Park et al.[44].

Comparing the PI3K-Akt pathway coverage on a phosphoproteome level, both the equal loading and microscaled TMT approach are able to cover almost the whole pathway, with only very few pathway members only covered by the equal loading TMT (Supplementary Fig. 9B).

Similarly, the KEGG pathway for NSCLC (Supplementary Fig. 10A) and the Ras signaling pathway (Supplementary Fig. 10B), which regulates cell growth and can activate several other signaling pathways like the PI3K-Akt pathway or the Raf/MAPK pathway[45], can also be largely covered on a phosphoproteome level by both the equal loading and microscaled TMT approach.

To ascertain the reproducibility of the microscaled TMT approach, we designed five TMT11 plexes out of consecutive 10 µm FFPE slices from eight patients (4 ADC and 4 SCC) so that the samples were randomly assigned to TMT channels 1–8, then two channels were left empty to avoid carryover from the internal standard in channel 11. The sample amounts were again 20 µg for each patient sample and 100× more material for the internal standard mix. We then performed deep global proteome and phosphoproteome analyses as with the miscroscaled TMT samples. The replicate plexes show an average of 8896 proteins quantified across all five plexes. We quantified >8100 proteins in all plexes requiring at least 80% valid values and >6500 proteins were quantified among all five plexes with no missing values at all (Supplementary Fig. 11A). The average reporter ion intensities of all plexes were highly correlated with Pearson correlation coefficients between 0.93 and 0.96 for both ADC and SCC samples ($p < 0.01$, Supplementary Fig. 11B).

On the phosphoproteome level, we were able to quantify 9686 phosphosites on average and in these five TMT plexes a reasonable coverage of almost 4000 quantified phosphosites

across 80% of all tumor samples can be achieved (Supplementary Fig. 11C). The average reporter ion intensities still showed a good correlation between all five plexes both for ADC and SCC samples ($p > 0.01$, Supplementary Fig. 11D).

It has been shown before that one limitation of the TMT quantification method is the rising number of missing values when integrating more and more plexes. This effect is already seen on the protein level but is even stronger on peptide level[46].

A clear advantage of the microscaled TMT approached presented here is that it provides in-depth coverage on proteome and phosphoproteome levels but can tolerate much lower input, hereby enabling comprehensive proteomic characterization of very small clinical specimens such as needle core biopsies. Our reproducibility analysis across five microscaled TMT experiments showed a high degree of reproducibility and only minor losses in proteome coverage across plexes.

**Applying microscaled TMT profiling to clinical FFPE biopsies.** To test the microscaled approach on clinical FFPE needle biopsies, an independent set of eight FFPE needle biopsies consisting of four ADC and four SCC cases was processed with the SDS-SP3 protocol. We could extract, on average, 79 µg of protein material per biopsy. As with the biopsy equivalents before, we labeled 20 µg per biopsy sample with TMT and added a 100× booster channel of a reference mix of ADC and SCC samples. In this experiment, 6800 proteins were quantified from eight FFPE needle biopsies with no missing values, excluding those only found in the boosting channel. The quantifications cover summed up reporter ion intensities of 5 orders of magnitude, including NSCLC-relevant proteins such as KEAP1, NKX2-1, and ERBB2 (Fig. 5a).

For the phosphoproteome, 90% of the TMT-labeled peptide material were used and we could reach a coverage of 5200 quantified phosphopeptides (Fig. 5b, Supplementary Data 7), which is a significant improvement from up to 2000 identified phosphopeptides in label-free single-shot LC-MS/MS analyses of FFPE lung needle biopsies (Supplementary Fig. 12A).

Overall, similar numbers of proteins were quantified in TMT experiments of biopsies and biopsy equivalents (7137 vs. 6792) with an overlap of 78% (Fig. 5c). This overlap is very satisfying since some variation between patients is to be expected. On the phosphoproteome level, fewer phosphosites were quantified in the TMT biopsy experiments than in the biopsy equivalents (5243 vs. 8798), which could be due to the slightly different cellular composition of biopsies and resected tissue samples (Fig. 5d, Supplementary Data 8).

**FFPE proteome/phosphoproteome coverage for LFQ and TMT methods.** Protein yields differ for different NSCLC FFPE tissue sample types, where one 10 µm slice of a resected tissue FFPE sample with a tumor area of ~150 mm² yields 150 µg protein on average and a needle biopsy FFPE sample with a tumor area of <5 mm² yields 70 µg protein (calculated from 3 × 10 µm slices). A proteome quantification coverage of 4400 proteins for FFPE resected tissues and 2000 proteins for FFPE biopsies can be expected from one standard LFQ LC-MS/MS run with 1 µg digested peptide injected (Supplementary Fig. 12B). From 100 µg peptide input material, around 6300 phosphosites could be identified in a label-free experiment from resected tissue and ~1700 phosphosites on average from biopsies. This coverage, combined with the comparatively short hands-on time required for the preparation of LFQ samples, makes this technique well suited for high-throughput studies of large cohorts. For a deeper, more comprehensive coverage, which is particularly relevant for integrative studies combining proteomic data with genomic data,

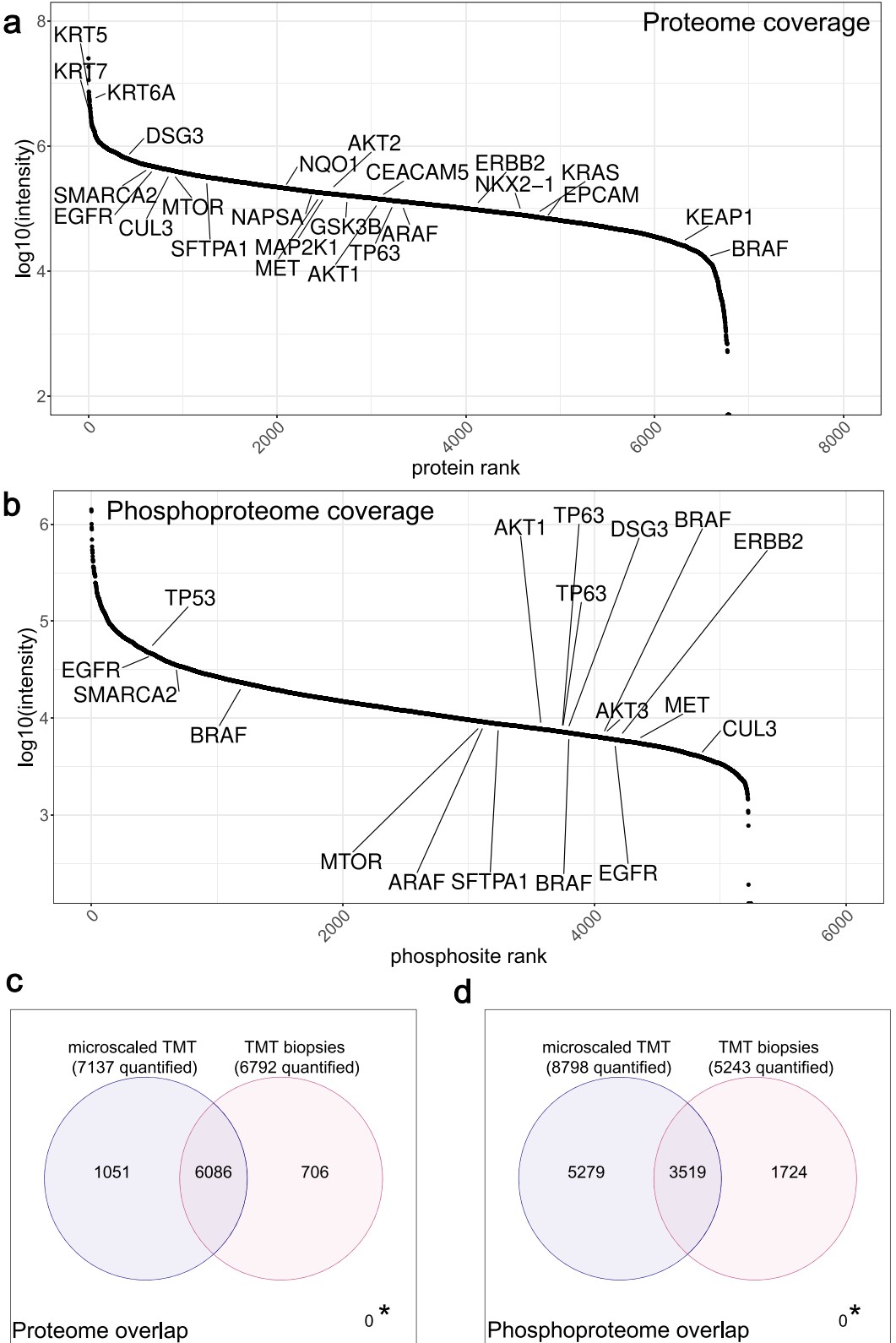

**Fig. 5 Deep proteome and phosphoproteome from FFPE needle biopsies. a** Log10 reporter ion intensity distribution over all proteins that were quantified in TMT biopsies. Source data are provided as a Source Data file. **b** Log10 reporter ion intensity distribution over all phosphosites that were quantified in TMT biopsies. Source data are provided as a Source Data file. **c** Venn diagram showing the overlap of quantified proteins between microscaled TMT and TMT biopsies. Only unique proteins and no isoforms were counted as quantified proteins. * marks proteins not quantified with either method (0). **d** Venn diagram showing the overlap of quantified phosphosites between microscaled TMT and TMT biopsies. * marks phosphosites not quantified with either method (0).

a multiplexed TMT approach is more suitable. By investing more efforts in sample preparation with labeling and offline fractionation, and instrument time, 8000 quantified proteins and over 14,000 quantified phosphosites can be reached with an equal loading TMT experiment for resected FFPE tissues. A microscaled TMT approach can boost the signal for FFPE needle biopsies to reach 6800 quantified proteins and 5200 quantified phosphosites (Supplementary Fig. 12B).

## Discussion

In the past, FFPE samples have been successfully used for several label-free proteomics studies in different types of thyroid cancer[47], fetal brain tissue[48], Alzheimer's disease brain tissue[49], colon carcinoma[9], uterine tissue[50], and lung tumors[17,51,52]. In our study, we show that the SDS-SP3 protocol performs best for protein extraction from lung cancer FFPE samples among three tested protocols. A similar comparison of FFPE sample preparation protocols has recently also been published by Griesser et al.[53] for lasercapture microdissected brain tissue FFPE samples, in which the SDS-SP3 protocol also provided the best results in terms of protein yield and proteome coverage. This indicates that the SDS-SP3 protocol has a wide range of possible applications across different laboratories and applications.

We applied the SDS-SP3 protocol to investigate the influence of pre-analytical variables both during routine pathology lab processing and proteomics sample preparation and found that commonly occurring formalin fixation time differences do not have a significant impact on the lung cancer proteome. An earlier report by Sprung et al.[7] also investigated the effect of different fixation times of one, two, and four days on the proteome coverage of colon adenocarcinoma FFPE samples. At a coverage of fewer than 500 proteins across all samples, the authors reported detrimental effects only after four days of fixation. Our study covers 24 and 72 h time intervals of fixation as they are frequently used for routine pathology lab samples and we found no detrimental effects on proteome coverage under these tested conditions. Regarding the length of incubation at 95 °C during lysis, a compromise has to be found between longer cooking time for increased protein decrosslinking and, therefore, proteome coverage and shorter cooking times for increased phosphoproteome coverage. In general, both proteins and phosphoproteins appear to be chemically quite heat-stable in FFPE samples.

Direct comparison of proteomes from fresh frozen vs. formalin-fixed specimens revealed no major difference in protein identifications[7,54] and minor or no effects on phosphoproteome quality for mouse liver tissues[11,54]. Our cell line experiments with freshly prepared formalin-fixed cells also revealed only minor differences in the proteome and phosphoproteome coverage compared to fresh frozen material. In contrast, a recent qualitative comparison of phosphoproteome coverage for fresh frozen vs. FFPE ovarian cancer specimens in long-term archived FFPE tumor specimens revealed a 75% lower identification rate of phosphopeptides, with the best reported FFPE tissue phosphoproteome coverage so far of 8000 phosphopeptides derived from 3000 proteins[13]. The improved protocols we report here, allow now to further improve the phosphoproteome coverage from FFPE cancer tissues to >14,000 phosphosites on 4400 proteins. This increase in coverage can be in part attributed to an automated IMAC enrichment on AssayMap tips that requires no desalting of the enriched phosphopeptides after elution from the IMAC resin, reducing overall sample losses. Particular caution needs to be applied to the interpretation of FFPE phosphoproteome data, since cold ischemia effects before complete fixation of the tissues may lead to activation of stress response pathways to some degree in the tissues[5]. Still, FFPE phosphoproteome

profiling can deliver valuable information by adding information on the phosphorylation status of proteins that were also detected in proteome profiles, by extending the overall list of detectable gene products by a class of proteins that were not covered in the corresponding proteome datasets and also by revealing activation events on proteins that show differential regulation across disease states only on the phosphorylation but not on the individual protein level. In our study, this was especially the case in the microscaled TMT approaches, where oncogenes and tumor suppressors such as Myc, FOXO3, and p53 were only quantified on the phosphoproteome level.

Our systematic comparison of commonly used FFPE protein extraction protocols, proteomics quantification methods, and input amount requirements on different FFPE sample types allows us to frame general guidelines for FFPE proteome analyses: LFQ requires only low amounts of input material for proteome analysis, relatively short sample preparation time and the least amount of measuring time per sample on the mass spectrometer (Fig. 6a). Therefore, LFQ is particularly well suited to study large cohorts of several hundred samples at moderate coverage. However, for phosphoproteome profiling, the approach is limited due to its relatively high starting amount requirements per individual sample. The equal loading TMT approach offers proteome and phosphoproteome at a much deeper coverage, but requires a relatively high starting amount of material and substantially more sample preparation time considering the additional sample processing steps for labeling, fractionation, and desalting. A clear advantage of the microscaled TMT approached presented here is that it provides in-depth coverage on proteome and phosphoproteome levels but can tolerate much lower input, hereby enabling comprehensive proteomic characterization of very small clinical specimens such as needle core biopsies (Fig. 6b).

Our reproducibility analysis across five microscaled TMT experiments showed a high degree of reproducibility and only minor losses in proteome coverage across plexes. Due to missing value propagation for low input samples across TMT cassettes, we recommend using the microscaled phosphoproteome approach only for up to four plexes, with up to 64 samples in TMT16-mode, until better methods with improved reporter ion sensitivity are developed. Previous cancer studies with 45 medulloblastoma cases[55], 27 breast cancer tumors[56] or 38 prostate cancer samples[57] and drug response profiling studies with 48 Jak2-mutated neoplasms[58] show that these cohort sizes can already be useful to molecularly characterize cancer subtypes and help in the discovery of future biomarkers.

Another label-free mass spectrometry approach for the analysis of large retrospective clinical cohorts, that was not explored in this study however, are data-independent acquisition (DIA) methods that were recently applied to quantify ~5000 proteins per sample from several FFPE tissues[14].

To conclude, our results show that that the commonly used quantification methods in mass spectrometry-driven proteomics, LFQ and TMT, both have different strengths and weaknesses for the analysis of FFPE tissues and offer versatile technological options to explore the consequences of genetic alterations in cancer on a functional proteome level.

## Methods

**Study design**. For the protein extraction protocol comparison, four replicates of FFPE lung ADC tissue samples containing one 10 μm scroll (ca. 150 mm² tumor area on average) each were used per protocol.

For the heat incubation time comparison, four replicates per time point were used for all three sample types (fresh-frozen cells, immediately formalin-fixed cells, and cells formalin-fixed after 1 h at 4 °C). The cells harvested from one T75 flask were used per replicate.

For the deep proteome and phosphoproteome analysis of ADC and SCC FFPE tissues, six 10 μm scrolls (ca. 150 mm² tumor area on average) were combined per

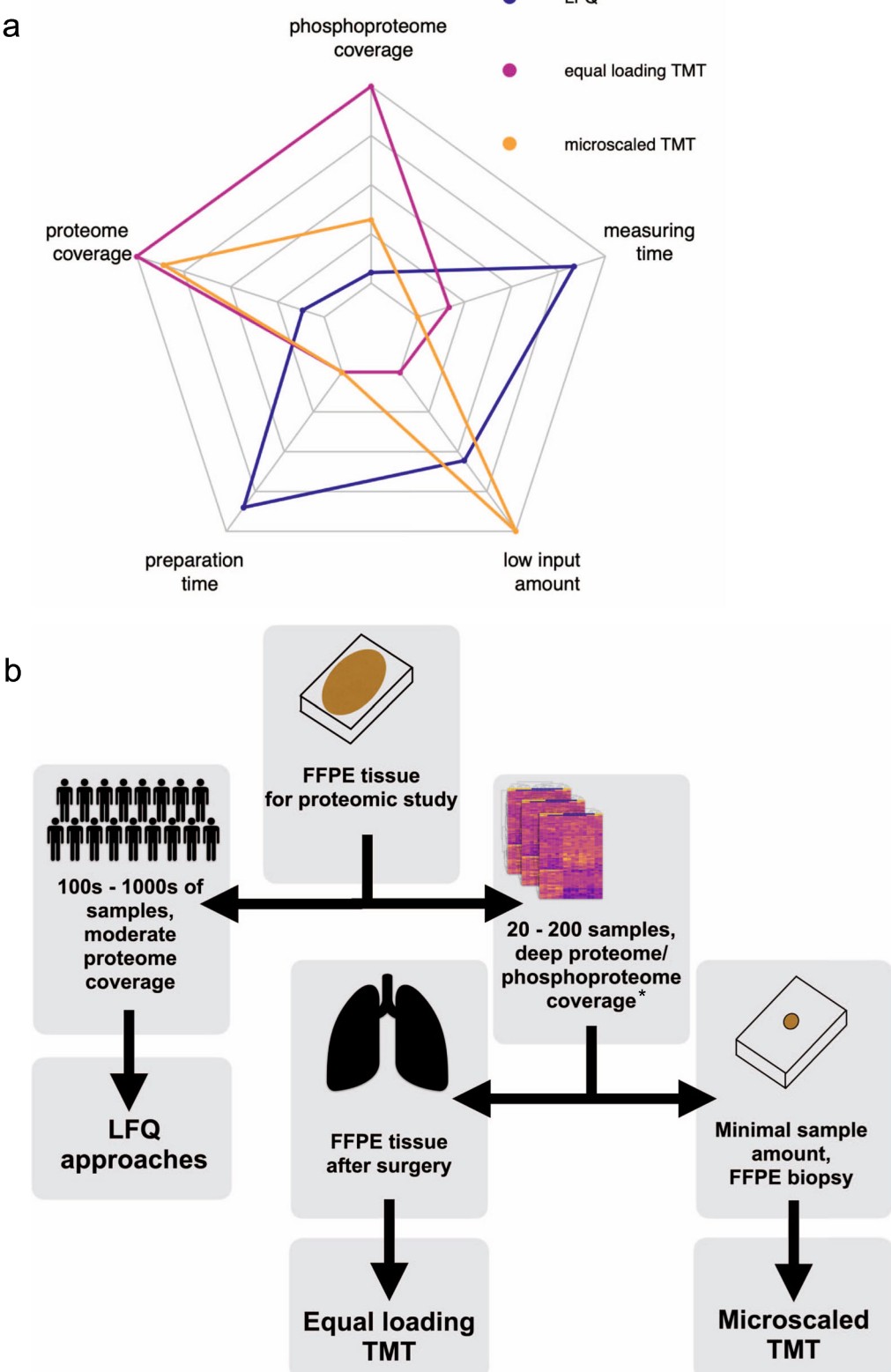

**Fig. 6 Guidelines for LFQ, equal loading TMT, and microscaled TMT proteome/phosphoproteome analysis in different settings. a** A spider plot showing the strengths and weaknesses of the three quantification methods presented in this study (LFQ in blue, equal loading TMT in magenta, and microscaled TMT in orange). From the inside to the outside the nodes represent quantities for the proteome between 1000 and 8000 proteins, for the phosphoproteome between 1000 and 14,000 phosphosites, for the mass spectrometry measuring time 7 and 1 h, for the low input material between 200 and 20 µg of total protein, and for the sample preparation time 50–10 h in the laboratory. For detailed values see supplementary Fig. 12B. **b** Flowchart for choosing a suitable quantification method for retrospective FFPE studies depending on the desired throughput and available sample amount. *Equal loading and microscaled TMT proteome analyses are recommended for 20–200 samples, microscaled TMT phosphoproteome analysis is recommended for 20–50 samples.

sample. From those, 200 µg peptide was used per sample for the equal loading TMT and 20 µg peptide per sample was used as "biopsy equivalents" for the microscaled TMT experiment.

Only for one out of the ten samples, the FFPE biopsy had enough material left after routine molecular pathology analyses to be included in the TMT biopsy experiment. Therefore seven additional FFPE needle biopsy samples from the archives were used for the TMT biopsy experiment. Three 10 µm slices (~5 mm² tumor area on average) were combined per biopsy sample.

FFPE NSCLC samples were acquired from the archive of the Institute of Pathology of the Charité University Hospital Berlin, Germany. Informed consent was obtained from all patients through the patient contract in accordance with institutional guidelines approved by the ethics board at the Charité Universitätsmedizin Berlin.

**Immunohistochemical staining**. Immunohistochemical (IHC) staining was performed on the BenchMark XT (Ventana) automated slide stainer for cytokeratins 5/6 (clone EP24, EP67 (abcam), diluted 1:100) and cytokeratin 7 (clone OV-TL12/30 (Dako), diluted 1:1000) according to the manufacturer's instructions.

**Deparaffinization**. For the resected tissue samples, 10 µm scrolls (ca. 150 mm² average tumor area) of lung FFPE tissues that had been routinely fixed in 4% buffered formalin were deparaffinized by three incubations with 1 mL xylene. After brief vortexing, the samples were centrifuged at 12,700 × g for 10 min and the supernatant was discarded. The samples were rehydrated afterwards by incubation with 1 mL 100%, 80%, and 70% ethanol, followed by an air-dry step for 10 min under a fume hood and stored at −80 °C until use.

For the FFPE biopsies, 10 µm slices (5 mm² tumor area on average) were mounted on microscope slides, incubated at 70 °C for 10 min to melt the paraffin, and cooled down to room temperature. Afterwards, the glass slides were incubated in xylene for 10 min, briefly air-dried, and then incubated in 100% ethanol for 10 min. The tumor area that had been marked by a pathologist before was then scraped off and transferred into a 1.5 mL tube using 50% ethanol. The samples were then centrifuged at 12,700 × g for 10 min, the ethanol was removed, and the samples were air-dried.

**Cell culture**. HEK293 cells were cultivated in 1% DMEM with 10% FCS and 1% penicillin-streptomycin. The cells were processed once they reached about 90% confluency.

After removal of medium and two washing steps with cold PBS, four T75 flasks of cells were fixed directly with formalin for three hours at room temperature. Afterwards, they were again washed with PBS and harvested.

Another four flasks were left in PBS at 4 °C for 1 h, then PBS was removed, they were fixed with formalin for 3 h and afterwards again washed with PBS and harvested.

A third batch was harvested and frozen directly.

**Direct trypsinization protocol (DTR)**. The samples were processed according to the protocol by Föll et al.[12]. Briefly, deparaffinized FFPE tissues were incubated with 100 µL buffer containing 0.1% Rapigest, 0.09 M HEPES, pH 8,0, 0.2 mM DTT in LC-MS grade water for 2 h at 95 °C in a thermoshaker. Afterwards, the samples were treated with a Bioruptor Pico sonicator (20 cycles on high, 30 s on/off) and centrifuged for 10 min at 12,700 × g. An aliquot of the supernatant was used for protein level BCA. After reduction and alkylation with 5 mM DTT and 15 mM IAA, the samples were pre-digested with trypsin and endopeptidase C (LysC) (enzyme:substrate ratio 1:50) for 2 h at 50 °C. The samples were then cooled to room temperature and incubated at 37 °C with fresh trypsin and LysC (enzyme:substrate ratio 1:50) overnight. The digested peptides were acidified with 100% formic acid the next morning, insoluble particles were spun down, and the peptides were desalted on C18 material. An aliquot of the desalted peptides was used for a peptide level BCA.

**Sodium deoxycholate-based protocol (SDC)**. The SDC protocol was first described by Wakabayashi et al.[11]. The deparaffinized tissues were incubated with 80 µL buffer containing 2% SDC and 1 mM EDTA in 50 mM Tris in a thermoshaker for 2 h at 95 °C, treated with a Bioruptor Pico sonicator (20 cycles on high, 30 s on/off), and centrifuged for 10 min at 127,000 × g. An aliquot of the supernatant was used for protein level BCA. Samples were reduced and alkylated with 12.5 mM DTT and 55 mM IAA and then diluted 1:5 with Tris and then predigested with LysC at 25 °C and digested overnight with trypsin at 37 °C (enzyme:substrate ratio 1:50). After digestion, ethyl acetate was used to remove the remaining paraffin, and the samples were acidified with 100% formic acid. Insoluble particles were spun down and the peptides were desalted on C18 material. An aliquot of the desalted peptides was used for a peptide level BCA.

**Sodium dodecyl sulfate-based protocol (SDS-SP3)**. The protocol was first described by Hughes et al.[10]. Here, the deparaffinized tissues were incubated with 40 µL buffer containing 50 mM Tris, 1% SDS, and 2 µL benzonase for 30 min at 37 °C. Afterwards, 40 µL buffer containing 2% SDS and 5 mM DTT were added and

the samples were incubated for 2 h at 95 °C, treated with a Bioruptor Pico sonicator (20 cycles on high, 30 s on/off) and centrifuged for 10 min at 127,000 × g. An aliquot of the supernatant was used for protein level BCA. After alkylation with 50 mM IAA, the remaining IAA was blocked with 113 mM DTT. 10 µL mix of paramagnetic beads containing 1:1 hydrophilic and hydrophobic beads were added for a concentration of 1 µg protein:10 µg beads. The concentration of organic solvent was increased to >70% by the addition of 800 µL acetonitrile and the proteins were incubated with the beads first on the bench and then in the magnetic rack. After the beads had settled, the supernatant was removed, and the beads were washed three times with 70% ethanol. Proteins were eluted from the beads with 100 µL 50 mM HEPES, pH 8. Digestion was performed overnight with LysC and trypsin (enzyme:substrate ratio 1:50). The next morning, samples were again incubated on the magnetic rack and the bead-free supernatant was transferred to fresh tubes and the beads were washed again with 100 µL HEPES. Samples were acidified with 100% formic acid, and desalted on C18 material. An aliquot of the desalted peptides was used for a peptide level BCA.

**TMT labeling and fractionation**. Dried down peptides (200 µg for equal loading TMT and 20 µg for the microscaled experiments) were reconstituted in 25 µL 50 mM HEPES, pH 8 for the patient samples selected for the deep multiplexed proteome analysis. The remaining 18 samples were combined in equal amounts for an internal standard, of which 200 µg were used for the equal loading TMT experiment and 2000 µg for the microscaled TMT experiments with biopsy equivalents and actual needles biopsies. The samples and the internal standard were labeled with TMT reagent with a reagent:peptide ratio of 1:1 for 1 h, then the reaction was quenched with 1 µL 1 M Tris for 15 min and the samples were mixed, dried down, and subsequently desalted on C18 material.

High pH reverse phase (hpH) fractionation of the labeled samples was performed on an Agilent 1290 system, as described by Mertins et al.[19]. Here, the samples were separated over a 96 min gradient with increasing acetonitrile concentration ranging from 2 to 54% acetonitrile with 2.5% ammonium formate.

The samples were initially separated into 96 fractions, of which every 28th fraction was then combined for the global proteome analysis. Ten percent of these 28 fractions were dried down and used for global proteome analysis. The remaining 90% were further combined into ten fractions for phosphoproteome analysis.

**Phosphopeptide enrichment**. Phosphopeptide enrichment was performed on an Agilent Bravo automated liquid handling platform using Fe(III)-NTA cartridges. Hundred micrograms of peptide were used as input material for the cooking time comparison and duplicates of 100 µg TMT-labeled and hpH-fractionated peptide were enriched and subsequently combined for the deep phosphoproteome analysis.

**Mass spectrometry data acquisition**. Data acquisition for LFQ comparison of ADC and SCC, as well as all TMT experiments, was performed on a Q Exactive HF-X instrument (Thermo Scientific, Xcalibur software version 4.1.31.9) coupled to an easy nLC 1200 system (Thermo Scientific). For the cooking time comparison, a Q Exactive Plus instrument (Thermo Scientific, Xcalibur software version 3.0.63) with an easy nLC 1200 system was used. The peptides were separated over a 110 min gradient with a flow rate of 250 nL with increasing concentration of buffer B (up to 60%) on a 20 cm reversed-phase column packed in-house with 1.9 µm beads (ReproSil Pur, Dr. Maisch GmbH).

The Q Exactive Plus was operated in data-dependent mode with 70 K MS1 resolution, AGC target of 3 × 10⁶ ions, and a maximum injection time of 50 ms, choosing the top 20 ions for MS2 scans with 17.5 K resolution, AGC target of 5 × 10⁴ ions and maximum injection time of 250 ms.

The Q Exactive HF-X was set up for 60 K MS1 resolution with an AGC target of 3 × 10⁶ ions and maximum injection time of 10 ms, followed by 20 MS2 scans with 45 K resolution, AGC target of 1 × 10⁵ ions, and maximum injection time of 86 ms.

**Data analysis**. Database searches were performed using MaxQuant (v1.6.3.3)[59] and the human reference proteome (UP000005640, downloaded 01/2019). Oxidation (M) and acetylation (N-term) were always included in the search as variable modifications, as well as carbamidomethyl cysteine as a fixed modification.

For the label-free comparison of 30 NSCLC cases, the integrated LFQ calculation was activated and the "match between runs" feature was used, as well as for the single-shot phosphopeptide-enriched samples, while it was turned off for protocol comparison and heat-incubation comparison. For TMT analyses, MS2 based labeling was chosen and the correction factor for each channel was entered according to the manufacturer's information. Reporter PIF was set to 0.5. For phosphoproteome analyses, phosphorylation (STY) was included as an additional variable modification.

**Variable modification settings for FFPE samples**. Resected tissues or tissue biopsies are fixed in a solution of 3.7% formaldehyde in water with methanol as a stabilizer, where they are dehydrated and stabilized by the formation of crosslinks between proteins[6]. Here, the formaldehyde reacts with primary amino groups of the proteins, which are then dehydrated and, in turn, react with amino acid residues within the protein itself or in other proteins[31]. After protein extraction and enzymatic digestion, both peptides that have been extracted and digested

completely and peptides that still contain the formaldehyde-induced modifications will likely be found, increasing the sample complexity and interfering with the confident peptide and protein identification[60,61]. N-terminal formylation[11], lysine methylation[45,] and methionine oxidation[62] have been reported, among others. Besides the standard MaxQuant settings for label-free analyses, formylation, the addition of a methylol group ($-CH_2O$), and dimethylation (KR) were set as additional modifications.

**Filtering of protein identifications and statistical analysis**. Proteins that were flagged as potential contaminants by MaxQuant were not removed during the analysis later on since three human cytokeratins are used as immunohistochemical markers for ADC and SCC. Reverse hits and proteins only identified by site were removed.

The limma R package[63] was used to perform two-sided moderated $t$-tests for the comparison of ADC and SCC in the different experiments. Multiple testing correction was performed via the Benjamini-Hochberg procedure and only hits with an adjusted $p$-value $< 0.05$ were considered statistically significant.

For the label-free proteome analyses of 30 FFPE samples, 16 ADC and 14 SCC samples were compared with two replicates each with a moderated $t$-test. The resulting -log10 of the adjusted $p$-values was then plotted over the log2 fold change between ADC and SCC in volcano plots. Proteins with a Benjamini-Hochberg-adjusted $p$-value $< 0.05$ were marked as statistically significant hits. The subset of 5 ADC and 5 SCC samples was analyzed in the same way.

Gene set enrichment analysis was performed with the ssGSEA and PTM-SEA tool for proteome and phosphoproteome analysis, respectively[37].

GO term enrichment was performed using the clusterProfiler R package[64] and only hits with Benjamini-Hochberg-adjusted $p$-values $< 0.05$ were considered statistically significant.

Pearson correlations were calculated and plotted using the corrplot R package[65]. KEGG pathway plots were rendered with the Pathview R package[66].

**Potential blood contamination quality control in needle biopsies**. Needle biopsies are taken during bronchoscopy or CT-guided procedures without the possibility to limit blood flow to the target tissue. Therefore the biopsies often contain more blood than resected tissue FFPE samples. This is not an issue for DNA analyses, since erythrocytes do not contain DNA, but they do contain proteins which in high abundance could influence the identification and quantification of tumor proteins. To investigate the difference in blood content in biopsies and biopsy equivalents, we used a list of 276 proteins identified in dried blood spots by Chambers et al.[67] as a reference. In the biopsy equivalents (derived from resection specimens), 199 proteins out of 276 were identified and 181 out of 276 were identified in the needle biopsies. The intensity distributions of blood proteins in both experiments behave very similarly to the majority of the proteins showing a summed-up intensity between $10^{10}$ and $10^{11}$ (Supplementary Fig. 13). Substantial differences between the proteomes of FFPE resected tissues and needle biopsies due to blood contamination can, therefore, be excluded.

**Reporting summary**. Further information on research design is available in the Nature Research Reporting Summary linked to this article.

## Data availability

Mass spectrometry raw files and processed MaxQuant datasets are available on the PRIDE proteomics data with the accession code PXD024800. Source data are provided with this paper.

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

## Acknowledgements

This work was supported by the German Ministry of Education and Research (BMBF), as part of the National Research Node "Mass spectrometry in Systems Medicine" (MSCoresys), under grant agreement 031L0220B to P.M. and 031L0220A to F.K., and also by the Deutsche Forschungsgemeinschaft (DFG – German Research Foundation) under grant agreement SFB 1449 "Dynamic Hydrogels" (projects C03, Z01) to P.M. We thank B. Meyer-Bartell for the excellent technical assistance.

## Author contributions

F.K., J.N., and S.S. collected clinical samples and sample information, C.F., M.H., S.N., and C.B. performed experiments; C.F. analyzed data and prepared figures; M.Z. provided input for statistical analysis; C.F., F.K., and P.M. wrote and revised the manuscript with input from all authors; C.F., F.K., M.K., and P.M. planned the experiments, conceived the microscaled TMT approach; F.K. and P.M. supervised the study.

## Funding

## Competing interests

The authors declare no competing interests.
