## [Peer Review File · Nature Communications]

REVIEWER COMMENTS

Reviewer #1 (Remarks to the Author):

Mertins Nature Communications

Given the potential utility of archived FFPE samples with substantial clinical follow-up data as specimens for retrospective studies correlating proteomic features with clinical outcomes, any advances in the routine use of these specimens for deep comprehensive and quantitative proteomic and phosphoproteomic analysis would represent significant contributions to the field. This manuscript is a substantial step toward that goal, in that it not only benchmarks several variables with the potential to affect analytical output (e.g., protein extraction procedures, formalin fixation time, duration of heat treatment, and time from tissue extraction to fixation), it also compares three different mass spectrometry approaches - label free quantification, equal loading TMT multiplexing, and boost channel TMT multiplexing (referred to as 'microscaling TMT'). In general, the manuscript is logically organized, clearly written, and results are appropriately interpreted in almost all cases (the exception being a slight over-interpretation of the GO results associated with Supplemental Figure 5B), while the work of others in the field is generally given appropriate credit. Adequate experimental and technical details are provided to allow others to reproduce the study, and adopt the recommended protocols.

The results presented clearly demonstrate the feasibility of using FFPE samples for reliable quantitative proteomic and phosphoproteomic profiling and establish generalizable guidelines for developing appropriate workflows for analysis of FFPE samples, depending on the needs of the experimental design. In this regard, the spider diagram in Figure 6 is particularly useful. Moreover, the demonstration that the boosting 'microscale TMT' approach can provide usefully deep coverage of both global and phospho- proteomes from FFPE needle biopsy samples is a significant advance for the field, offering a substantial expansion in the types of archived samples available for comprehensive proteomic profiling. The results in Figure 4 clearly show a significant increase in the depth of phosphoproteome coverage compared to prior efforts, such as Piehowski et al., and the attribution of this improvement to automated IMAC analysis on AssayMap tips that avoid the need for desalting is reasonable, although not experimentally validated.

The one weakness of this manuscript is the failure to adequately address the lack of overlap between phosphosites identified by microscaling TMT compared to equal TMT in Figure 4C; 8,167 phosphosites is a substantial loss, when going from 200 ug loading to 20 ug loading – and a gain of 2832 phosphosites in the microscaled samples compared to the equal TMT is equally troubling. One would like to know the difference in physical chemistry between non-overlapping peptides in each

method that might account for the observed differences; trying to attribute differences this large to patient heterogeneity is disingenuous. In addition to wondering about the physical attributes underlying the lack of overlap, one is also very curious about the biology that is missed with each approach, compared to the other. In this regard, a Figure similar to Figure 3A, but with different colors for the phosphopeptides identified in equal TMT versus microscaled TMT would have been quite informative. Although Supplemental Figure 6 appears to contain the PI3K-AKT pathway information from the microscale analysis, it is too difficult to compare these two figures manually, and actually identify what has been lost; a single composite figure would have been better. Furthermore, some indication of coverage in additional pathways, beyond PI3K-AKT would have been useful – for example, the EGFR-RAS pathway. Widespread adoption of the microscaling TMT approach for phosphoproteomic analysis of needle biopsies will require substantial evidence that no significant biological insights are lost in the process, or at the very least a clear basis for predicting which biological insights are most likely to be lost with microscaling.

Reviewer #2 (Remarks to the Author):

Authors in this manuscript provide comprehensive review of three methods of FFPE samples extraction and analysis for both regular and phospho-proteome samples. The optimization of protein de-crossing procedure is carefully discussed as well.

Three main procedures of sample extraction including: DTR, SCD, and SCD-SP3 are discussed, as well as analytical methods such as label-free and TMT based workflows. Micro sampling with TMT boosting channel was applied for cases of samples which were too limited for analysis based on standard needle based biopsy.

Authors clearly show the advantage of SCD-SP3 type sample preparation protocol followed by TMT labeling strategy. Application of this strategy greatly aids with boosting channel intensity for samples where only a limited amount may be procured.

There are couple of technical questions that needs to be addressed in the review:

1. In many figures for differentially expressed protein. cytokeratins were shown to have the most amount of change and were also the most abundant proteins according to abundance figure as Suppl. Fig 5. Unless authors have a solid proof that those proteins are coming from the sample and were not typical laboratory contaminants during the sample prep procedure, I would recommend to remove those from expression analysis.

Next technical question is to author's statement that phospho-proteome is not changing upon fixation. The current manuscript experiment shows only differences between time points of cell fixation, but not a comparison to fresh frozen cells. It would be nice to show a correlation of phospho-proteome of the best optimized authors protocol for cross linked cell based material vs

freshly frozen cells together with enzyme inhibitors, then correlations among those two phosphoproteomes would be able to be run within the same TMT experiment and thus could be used to support such statement.

Overall authors show great progress in analysis of FFPE type samples, especially for case of samples with low amounts from needle biopsy with TMT channel boosting.

Reviewer #3 (Remarks to the Author):

Summary: In this paper Dr. Mertins and colleagues describe a comparative analysis of three methods to extract and process proteins and phosphopeptides from FFPE fixed samples. After selecting an optimal method among the methods tested the authors evaluate the robustness of the method to a range of fixing conditions and other parameters and find the procedure to be relatively robust. The authors then demonstrate that the method is compatible with needle biopsies. The samples used are two classes of NSCLC samples (ADC and SCC).

General comments: The study addresses an important issue in clinical proteomics, the analysis of proteins and phosphoproteins from FFPE samples by mass spectrometry. The experiments are generally well described and competently done and the data analysis is based on standard methods. The degree of novelty of the study is low. The tested methods have been described and to some extent compared before with similar conclusions (SDS-SP3 superior); the mass spectrometric methods, including the use of a carrier TMT channel are standard as is the method to enrich phosphopeptides. There are no clinically relevant insights generated. The clinical arguments are limited to stating that some of the expected differences between the sample classes are detected in the data. No attempt is made to explain some of the newly detected proteomic or phosphoproteomic differences beyond some GO annotation. From the purely technical side the paper therefore largely confirms prior results and provides no fundamentally new insights.

Perhaps the conceptually most significant weakness of the paper is the singular focus on numbers or peptides/proteins/phosphopeptides that are identified. The study is positioned in the arena of clinical proteomics and while high coverage is desirable, it is by no means sufficient to establish a method as useful. Clinical studies require the analysis of rather large sample cohorts at a high degree of reproducibility to deal with variability and confounding effects and the study makes no attempt to distinguish technical from biological variability of the methods. It would be essential to demonstrate the overlap/missing value distribution and quantitative variance of control sample sets across minimally a few (e.g. five TMT sets). In fact results shown in supp Fig.2 are disconcerting because it

appears that with a higher number of samples compared in a volcano plot the number of differentially abundant proteins increases, a patterns that suggests a high degree of variability.

Overall, the paper has very minor technical or conceptual novelty, and critical issues for the intended field of application are not addressed.

Specific comments:

1) Supp Fig 2. The authors are requested to describe how the volcano plots were constructed and the results need to be described in more detail. Specifically, the authors need to describe how the data from the different samples in a class were combined, the variability of the same proteins from the same sample class and what fraction of proteins detected as differential or non differential were directly measured and identified by pattern matching respectively. Generally, the reader should be able to better assess, preferably with additional control data to what extent the observed differences are biological as opposed to technical.

2) The selection of the optimal protocol is not well enough documented. What amount of sample was processed, what is the reproducibility and variability of the data as a function of sample size as the stated goal is needle biopsy level samples.

3) Investigation of fixation time. The number of proteins identified is not really an informative metric. The authors should show a volcano plot to detect potential biases affecting specific protein sets for the conditions compared.

4) Generally the paper lacks indications on the amounts of sample required and processed. E.g. dimension of needle biopsy samples actually used and not just "needle biopsy equivalents".

5) Biopsy equivalent --- did they use biopsies or not and for what and with what results? What was actually done for phosphopeptide analysis from needle biopsies? Was the remaining ca. 60 microgram of peptide sample subjected to IMAC enrichment?

6) Generally the number of replicates are too low and too diverse to support confident conclusions. It is recommended to compare replicates of very similar samples e.g. biopsy level samples from the same resected tumor area to determine the technical variability.

7) The interpretation of the detected differential molecules between ADC and SCC samples is superficial. In addition of highlighting proteins detected as differential that are known as differential in the literature the authors also should describe proteins that are expected to change but were not detected as changed and some effort should be made to assess the differential molecules that are not yet in the literature in this scenario. E.g are these likely genuine differences (e.g. related to known biochemical differences between the samples) or are they artifacts e.g different levels of blood proteins or rather likely contaminants?

REVIEWER COMMENTS

Reviewer #1 (Remarks to the Author):

Mertins Nature Communications

Given the potential utility of archived FFPE samples with substantial clinical follow-up data as specimens for retrospective studies correlating proteomic features with clinical outcomes, any advances in the routine use of these specimens for deep comprehensive and quantitative proteomic and phosphoproteomic analysis would represent significant contributions to the field. This manuscript is a substantial step toward that goal, in that it not only benchmarks several variables with the potential to affect analytical output (e.g., protein extraction procedures, formalin fixation time, duration of heat treatment, and time from tissue extraction to fixation), it also compares three different mass spectrometry approaches - label free quantification, equal loading TMT multiplexing, and boost channel TMT multiplexing (referred to as 'microscaling TMT'). In general, the manuscript is logically organized, clearly written, and results are appropriately interpreted in almost all cases (the exception being a slight over-interpretation of the GO results associated with Supplemental Figure 5B), while the work of others in the field is generally given appropriate credit. Adequate experimental and technical details are provided to allow others to reproduce the study, and adopt the recommended protocols.

The results presented clearly demonstrate the feasibility of using FFPE samples for reliable quantitative proteomic and phosphoproteomic profiling and establish generalizable guidelines for developing appropriate workflows for analysis of FFPE samples, depending on the needs of the experimental design. In this regard, the spider diagram in Figure 6 is particularly useful. Moreover, the demonstration that the boosting 'microscale TMT' approach can provide usefully deep coverage of both global and phospho- proteomes from FFPE needle biopsy samples is a significant advance for the field, offering a substantial expansion in the types of archived samples available for comprehensive proteomic profiling. The results in Figure 4 clearly show a significant increase in the depth of phosphoproteome coverage compared to prior efforts, such as Piehowski et al., and the attribution of this improvement to automated IMAC analysis on AssayMap tips that avoid the need for desalting is reasonable, although not experimentally validated.

The one weakness of this manuscript is the failure to adequately address the lack of overlap between phosphosites identified by microscaling TMT compared to equal TMT in Figure 4C; 8,167 phosphosites is a substantial loss, when going from 200 ug loading to 20 ug loading – and a gain of 2832 phosphosites in the microscaled samples compared to the equal TMT is equally troubling. One would like to know the difference in physical chemistry between non-overlapping peptides in each method that might account for the observed differences; trying to attribute differences this large to patient heterogeneity is disingenuous.

Response: We thank the reviewer for the supporting comments on our comprehensive FFPE proteome/phosphoproteome profiling approach. We agree that it is non-intuitive to judge with the information given in the original Figure 4C why there is a difference in quantified phosphosites between the equal loading TMT with 200 µg to microscaled TMT with 20 µg input per sample. We attribute these differences to multiple different contributing factors that we have now further described in the manuscript with additional text and figures (see below).

1. Overall phosphosites identifications differ between equal and microscaled TMT experiments due to different sample nature – 16,492 vs 13,849 phosphosites (83.9%), respectively. Due to the large sample amount needed for the internal standard we had to mix samples from 20 NSCLC cases to generate the internal standard. While the multiplex equal loading sample is equally comprised of 10 tumor samples (with 200 µg loading for each), the microscaled sample consists to >90% of an equally mixed internal standard samples (2,000 µg; mix of 20 tumors) and an additional 8 tumor samples (20 µg each). The higher complexity of the internal standard coincides with an overall reduced rate of identifications. On the proteome level this reduction is down to 90.5% (7979/8815 proteins). The different sample nature may also explain the exclusive quantification of 2,832 phosphosites in the microscaled but not the equal loading TMT experiment.
2. In the microscaled TMT experiment we observe a 10.9% decrease going from identified to quantified phosphosites, whereas there is only a 0.75% difference in the equal loading TMT (see new Suppl. Fig 8A/8B and calculations below). These dropouts are caused by loss of signal in the tumor TMT channels, while a signal is still present in the internal standard TMT channel.
3. To evaluate the degree of variation that can be expected for the two methods, we have compared the phosphosites quantified in the two replicate plexes for both equal loading and microscaled TMT (see Suppl. Fig 8A/8B) and we observed that the variation between replicates is higher in the microscaled TMT approach with 22.2 % less phosphopeptides that were reproducibly quantified between replicates.

We have added the following section on page 14/15 to explain the reduced phosphosite quantification numbers of the TMT microscaling approach:

“The observed coverage differences between equal loading and microscaled TMT phosphoproteomes can be attributed to multiple factors. First, the increased complexity of internal standard containing microscaled TMT samples coincides with decreased identifications on protein and phosphosite level by 9.4% and 16%, respectively (Suppl. Fig. 8A/B). Due to higher sample amount requirements for the internal standard a mix of 20 different NSCLC tumors was generated, which is more complex than the individual 10 tumors in the equal loading experiment. Second, in the microscaled TMT experiment we observed a 10.9% decrease going from identified to quantified phosphosites, whereas there is only a 0.75% difference in the equal loading TMT. These dropouts are caused by loss of signal in the tumor TMT channels, while a signal is still present in the internal standard TMT channel (100x more input). Third, comparing the two replicate plexes for each TMT experiment (px1 and px2 for equal loading TMT and px3 and px4 for microscaled TMT), the percentage of phosphosites reproducibly quantified in both plexes is higher in the equal loading TMT than in the microscaled TMT (85.7% vs 63.5%, Suppl. Fig. 8A/8B). The 22.2 percentage points difference are likely due to the higher variability of lower abundant phosphopeptides in microscaled TMT.”

Please see below a summary table of the phosphosite numbers reported in the new Suppl. Fig. 8A/B:

	equal	%	micro	%
replicate 1 (blue)	882 (px1)	5.4%	2,269 (px3)	16.4%
overlap	14,133	85.7%	8,798	63.5%
replicate 2 (magenta)	1,353 (px2)	8.2%	1,270 (px4)	9.2%
Not quantified	124	0.75%	1,512	10.9%

Total identified	16,492		13,849	
Difference due to different sample nature is 16% (1-13,849/16,492)				
Delta in reproducibly quantified p-sites is 85.7-63.5=22.2%				

As requested we also added a comparison of the hydrophobicity index (see Krokhin et al. 2009) as an example for physical-chemical properties for all phosphopeptides and non-phosphorylated peptides in both datasets which show very similar distributions over the hydrophobicity range (see new Suppl. Fig 8C). We also show the hydrophobicity index for those phosphopeptides either quantified in equal loading or microscaled TMT approaches alone (the parts of the Venn diagram that do not overlap; Suppl. Fig. 8D) and they also show a very similar intensity distribution. This suggests that the differences in quantified phosphosites are not due to a hydrophobicity bias in the microscaled TMT experiment, but more likely due to the higher variability on (phospho-) peptide level and the more challenging experimental design. Please see also addition to manuscript page 15: *“The hydrophobicity index¹ for the phosphopeptides quantified in equal loading and microscaled TMT is distributed very similarly, both for the total phosphoproteome (Suppl. Fig. 8C) and the subset of phosphopeptides that was either uniquely quantified in the equal loading TMT dataset or in the microscaled TMT dataset (Suppl. Fig. 8D, see also Venn diagram Fig. 4C). This suggests that the two TMT approaches do not show a bias with respect to the type of phosphopeptides that are enriched and subsequently quantified, but that the differences arise merely from the variability that is known from data-dependent acquisition analysis of low abundant phosphopeptide samples.”*

In addition to wondering about the physical attributes underlying the lack of overlap, one is also very curious about the biology that is missed with each approach, compared to the other. In this regard, a Figure similar to Figure 3A, but with different colors for the phosphopeptides identified in equal TMT versus microscaled TMT would have been quite informative.

Response: Thank you for this suggestion. To make the pathway easier to compare for the two methods, we have now included a figure comparing the coverage of the PI3K-Akt pathway on a phosphoproteome level for equal loading TMT and microscaled TMT in Suppl. Fig. 9B, and added the following text to the manuscript on page 15:

“Comparing the PI3K-Akt pathway coverage on a phosphoproteome level, both the equal loading and microscaled TMT approach covered almost the whole pathway, with only very few pathway members occurring only in the equal loading TMT data (Suppl. Fig. 9B).”

Although Supplemental Figure 6 appears to contain the PI3K-AKT pathway information from the microscale analysis, it is too difficult to compare these two figures manually, and actually identify what has been lost; a single composite figure would have been better.

Response: Please see the comment above for the revised pathway figure which we believe is now clearer.

Furthermore, some indication of coverage in additional pathways, beyond PI3K-AKT would have been useful – for example, the EGFR-RAS pathway. Widespread adoption

of the microscaling TMT approach for phosphoproteomic analysis of needle biopsies will require substantial evidence that no significant biological insights are lost in the process, or at the very least a clear basis for predicting which biological insights are most likely to be lost with microscaling.

Response: Thank you for the comment on showing the coverage of additional pathways. We find the coverage of relevant oncogenic signaling pathways to be also sufficient with the microscaled TMT methods. We have now included a new figure (Suppl. Fig. 10A) depicting the coverage of the KEGG pathway “non-small cell lung cancer” coverage on a phosphoproteome level comparing equal loading TMT and microscaled TMT where signaling from EGFR to Ras and the MAPK signalling pathway is depicted.

We have also added a figure of the Ras signalling pathway coverage on the phosphoproteome level comparing equal loading and microscaled TMT approaches as Suppl. Fig. 10B.

Suppl. Fig. 10A/B are now described in the manuscript on page 15: “*Similarly, the KEGG pathway for non-small cell lung cancer (Suppl. Fig. 10A) and the Ras signaling pathway (Suppl. Fig. 10B), which regulates cell growth and can activate several other signaling pathways like the PI3K-Akt pathway or the Raf/MAPK pathway², can also be largely covered on a phosphoproteome level by both the equal loading and microscaled TMT approach.*”

Reviewer #2 (Remarks to the Author):

Authors in this manuscript provide comprehensive review of three methods of FFPE samples extraction and analysis for both regular and phospho-proteome samples. The optimization of protein de-crossing procedure is carefully discussed as well.

Three main procedures of sample extraction including: DTR, SCD, and SCD-SP3 are discussed, as well as analytical methods such as label-free and TMT based workflows. Micro sampling with TMT boosting channel was applied for cases of samples which were too limited for analysis based on standard needle based biopsy.

Authors clearly show the advantage of SCD-SP3 type sample preparation protocol followed by TMT labeling strategy. Application of this strategy greatly aids with boosting channel intensity for samples where only a limited amount may be procured. There are couple of technical questions that needs to be addressed in the review:

1. In many figures for differentially expressed protein. cytokeratins were shown to have the most amount of change and were also the most abundant proteins according to abundance figure as Suppl. Fig 5. Unless authors have a solid proof that those proteins are coming from the sample and were not typical laboratory contaminants during the sample prep procedure, I would recommend to remove those from expression analysis.

Response: We appreciate the reviewer’s comments and understand that certain lung cancer marker proteins such as cytokeratins need further explanations. We have included cytokeratins 5/6 and 7 in the analysis because they are used as diagnostic markers by pathologists to differentiate between lung squamous cell and adenocarcinoma. We have now added scans of immunohistochemical stainings for those keratins in Suppl. Figs 1A and 1B to show the tissue specificity of those markers. (See also manuscript page 6: “*immunohistochemical markers, like cytokeratins (KRT5/6/7, immunohistochemistry (IHC) stainings for ADC/SCC shown in Suppl. Fig. 1A and B),*”) and as confirmation that the keratins were not introduced during sample processing.

Next technical question is to author's statement that phospho-proteome is not changing upon fixation. The current manuscript experiment shows only differences between time points of cell fixation, but not a comparison to fresh frozen cells. It would be nice to show a correlation of phospho-proteome of the best optimized authors protocol for cross linked cell based material vs freshly frozen cells together with

enzyme inhibitors, then correlations among those two phospho-proteomes would be able to be run within the same TMT experiment and thus could be used to support such statement.

Response: Our submitted manuscript already included a label-free proteome and phosphoproteome comparison for both fresh-frozen and formalin-fixed cells in the heat incubation time course experiment. Please see manuscript page 9/10: *“Next, we investigated the impact of heat on the efficiency of the de-crosslinking and the stability of the proteome and phosphoproteome. To avoid sample processing conditions that would lead to protein degradation or loss of phosphorylation, we compared different heat incubation times in a cell culture experiment. HEK293 cells were either fresh-frozen directly after harvesting, formalin-fixed immediately on the plate, and then harvested or first incubated at 4°C for an hour and then formalin-fixed to mimic processing delays in the routine pathological lab. All samples were treated with the same SDS-lysis buffer and replicates of each type of sample were incubated either 10 min, 30 min, 60 min, or 120 min at 95°C (Suppl. Fig. 4A), cleaned up with SP3, digested and run in single-shot analyses. [...]*

Equal amounts of each sample (100 µg) were enriched for phosphopeptides by immobilized metal affinity chromatography (IMAC) and in all three types of samples, the lowest number of phosphopeptides was identified after 10 min of cooking at 95°C. Identification rates increased up to 50% with the maximum number of phosphopeptides identified at 60 min, then the rates decreased again slightly (Suppl. Fig. 4C). Noteworthy, phosphorylation modifications on proteins are stable enough to withstand heat incubation times of up to 2h at 95°C without any major abundance losses.”

As requested by the reviewer, we have now also performed a TMT comparison of fresh-frozen and formalin-fixed HEK293 cells and added new plots to show the pearson correlations of both sets of samples on proteome (Suppl. Fig. 5A) and phosphoproteome level (Suppl. Fig. 5B). See also manuscript page 10 for the description of the new results: *“Additionally, a TMT comparison of 5 replicates of fresh-frozen HEK293 cells and 5 replicates of formalin-fixed HEK293 cells showed a high correlation between the proteomes of fresh and formalin-fixed cells (Suppl. Fig. 5A) with Pearson correlation coefficients of 0.96 – 0.99 ($p < 0.01$). The phosphoproteomes of fresh-frozen and formalin-fixed cells also showed a good correlation with Pearson correlation coefficients of 0.7-0.9 ($p < 0.01$), an expected trend, considering that the variability is generally higher on the (phospho-)peptide level (Suppl. Fig. 5B).”*

Overall authors show great progress in analysis of FFPE type samples, especially for case of samples with low amounts from needle biopsy with TMT channel boosting.

Response: We thank the reviewer for the constructive criticism and overall support for our manuscript.

Reviewer #3 (Remarks to the Author):

Summary: In this paper Dr. Mertins and colleagues describe a comparative analysis of three methods to extract and process proteins and phosphopeptides from FFPE fixed samples. After selecting an optimal method among the methods tested the authors evaluate the robustness of the method to a range of fixing conditions and other parameters and find the procedure to be relatively robust. The authors then demonstrate that the method is compatible with needle biopsies. The samples used are two classes of NSCLC samples (ADC and SCC).

General comments: The study addresses an important issue in clinical proteomics, the analysis of proteins and phosphoproteins from FFPE samples by mass spectrometry.

The experiments are generally well described and competently done and the data analysis is based on standard methods. The degree of novelty of the study is low. The tested methods have been described and to some extent compared before with similar conclusions (SDS-SP3 superior); the mass spectrometric methods, including the use of a carrier TMT channel are standard as is the method to enrich phosphopeptides.

Response: We thank the reviewer for the valuable comments that have further improved our study (see below). At the same time, we respectfully disagree with the reviewer about the lack of novelty for our technical advancements of FFPE proteomics methods. While individual components of these methods have been used to study cancer in general, with proteomics before, we here provide valuable comparisons and guidelines that have not been described for the - as all reviewers agree - very important discipline of FFPE tissue proteomics so far. In particular, a FFPE SDS-SP3 protocol comparison has been only described in one study by Griesser et al. (2020) in a different context of FFPE laser capture microdissection. In our study, we compared the SDS-SP3 protocol for the more frequently studied analysis of macrodissected FFPE tissues and see strong precision differences to the other tested methods (see point 2). Booster TMT channels have been used before for single cell proteomics analysis (Budnik et al, Genome Biol 2018) or the analysis of fresh frozen leukemia cell lines or pancreatic islets (Yi et al, Anal. Chem 2019). Our study, however, is the first to systematically evaluate the pros and cons of using a TMT booster channel approach for a robust and deep coverage proteome/phosphoproteome profiling of FFPE tumor tissues. In addition, our FFPE proteomics methods yield among the highest reported proteome coverages >8,000 proteins and doubles the best reported FFPE phosphoproteome coverages with >14,000 phosphosites quantified at good reproducibility in the equal loading TMT mode.

There are no clinically relevant insights generated. The clinical arguments are limited to stating that some of the expected differences between the sample classes are detected in the data. No attempt is made to explain some of the newly detected proteomic or phosphoproteomic differences beyond some GO annotation. From the purely technical side the paper therefore largely confirms prior results and provides no fundamentally new insights.

Response: Our study has a methodological focus with lung cancer subtype comparisons as a use case to test how known and newly emerging disease effects can be studied in archived patient samples. Please see also response to point 7 below, where we provide biological insight. Also, as detailed in the answer above, a systematic technical comparison of the different methods we evaluate has not done before and an FFPE proteomics application guideline as presented in our manuscript, has also not been available before.

Perhaps the conceptually most significant weakness of the paper is the singular focus on numbers or peptides/proteins/phosphopeptides that are identified. The study is positioned in the arena of clinical proteomics and while high coverage is desirable, it is by no means sufficient to establish a method as useful.

Response: We agree with the reviewer that further analysis for the reproducibility and precision of the described FFPE proteomics methods were necessary, which we provide now in our response to the specific comments (see points 2 and 6 below).

Clinical studies require the analysis of rather large sample cohorts at a high degree of reproducibility to deal with variability and confounding effects and the study makes no attempt to distinguish technical from biological variability of the methods. It would be

essential to demonstrate the overlap/missing value distribution and quantitative variance of control sample sets across minimally a few (e.g. five TMT sets).

Response: We have added the requested experiments to the study. Please see response to point 6.

In fact results shown in supp Fig.2 are disconcerting because it appears that with a higher number of samples compared in a volcano plot the number of differentially abundant proteins increases, a patterns that suggests a high degree of variability.

Response: We think the results shown in Suppl. Fig. 2 behave as expected. Larger sample size always leads to higher statistical power, e.g. in t-tests the variance is divided by the sample size. Thus, for a constant group variance higher sample sizes will lead to better significances, and therefore at a constant significance level more differences can be statistically distinguished.

Overall, the paper has very minor technical or conceptual novelty, and critical issues for the intended field of application are not addressed.

Response: We think that our study fills an important methodological gap for the deep and robust proteome/phosphoproteome profiling of medium sized clinical FFPE cancer cohorts with a focused research question. As indicated by our FFPE proteomics guidelines in Figure 6 we recommend to use label-free proteomics analyses for large clinical sample cohorts of 100s to 1,000s of samples, to identify protein biomarkers or signatures in medium coverage proteomes. The equal loading and microscaled TMT FFPE proteomics methods described by us provide solutions for different clinical study designs in which very specific research questions are addressed in medium sized FFPE cohorts at deep proteome coverage. Many examples for such medium sized cancer proteomic/phosphoproteomic studies have been published recently with fresh frozen material focusing on cancer subtype comparisons for better disease understanding (Archer et al, Cancer Cell 2018; 45 Medulloblastoma cases) (Djomehri et al, Nature Comm. 2020; 27 metaplastic breast tumors/controls) (Latonen et al, Nature Comm. 2018; 38 prostate cancers/controls) or drug response profiling (Jayavelu et al, Nature 2020; 48 Jak2-mutated neoplasms) among many others. Our proposed methods now also enable similar studies with FFPE tissue cohorts.

Specific comments:

1) Supp Fig 2. The authors are requested to describe how the volcano plots were constructed and the results need to be described in more detail. Specifically, the authors need to describe how the data from the different samples in a class were combined, the variability of the same proteins from the same sample class and what fraction of proteins detected as differential or non differential were directly measured and identified by pattern matching respectively. Generally, the reader should be able to better assess, preferably with additional control data to what extent the observed differences are biological as opposed to technical.

Response: We apologize if the representation of the data was unclear. The volcano plots were constructed with the results of a moderated t-test (limma R package) comparison of a 16 by 14 or a 5 by 5 subset comparison of the ADC and SCC sample groups in subpanels A and B, respectively. We only used proteins with no missing values in at least one group for the calculations. Proteins with a Benjamini-Hochberg adjusted p value <0.05 were considered statistically significant and are highlighted in orange.

We have now edited the methods section to clarify this. Please see manuscript page 29: *“For the label-free proteome analyses of 30 FFPE samples, 16 ADC and 14 SCC samples were measured with two replicates each and ADC vs SCC groups were compared with a moderated t-test. The resulting -log₁₀ of the adjusted p-values was then plotted over the log₂ fold change between ADC and SCC in volcano plots. Proteins with a Benjamini-Hochberg-adjusted p-value <0.05 were marked as statistically significant hits. The subset of 5 ADC and 5 SCC samples was analysed in the same way.”*

As per your suggestion, we have added a clarification about the “match between runs” feature of MaxQuant that we used for the label-free samples and performed additional data analyses to study the effects of MBR on protein quantification in our study: We determined for all quantified proteins the fraction that was identified by MS/MS spectra and the fraction identified only with the MBR feature (see new Suppl. Fig. 3A). The vast majority of quantified proteins was identified by MS2 spectra (90%) and in this case the MBR algorithm only contributed a small part of lower intensity proteins (10%). This can be nicely illustrated on protein level when comparing individual ADC and SCC samples (see new Suppl. Fig. 3B and 3C). Proteins identified only by matching tend to have lower LFQ intensities and a higher degree of variability (average $r = 0.580$ vs 0.905) as shown in scatter plots of two representative ADC or SCC replicates. We included plots for all quantified proteins, for the significantly expressed proteins and also the NSCLC-markers that we used as quality controls in this study. Our data shows that variation between ADC and SCC can be assumed to be biological, as technical correlation is very high. Pearson correlations of technical replicate pairs for ADC and SCC were 0.995 and 0.994 (both $p < 0.01$), respectively. R of biological replicate comparisons for ADC and SCC samples were 0.888 and 0.917 on average.

Please see also manuscript page 7/8: *“The “match between runs” (MBR) feature of the MaxQuant platform³ was used to maximize the number of quantified proteins in the LFQ experiment. To ensure that the majority of quantified proteins is still identified from MS/MS spectra, we compared the log₂-transformed LFQ intensity distributions of all quantified proteins to the subset of proteins with MS/MS spectra-based identifications and those only identified with the MBR feature (Suppl. Fig. 3A). The vast majority of quantified proteins was identified via MS/MS spectra (90%) and in this comparison the MBR algorithm contributes only a small part of lower-intensity proteins (10%). This can also be seen in the individual ADC (Suppl. Fig. 3B) and SCC (Suppl. Fig. 3C) technical replicates where proteins identified only by spectral matching tend to have lower LFQ intensities and a higher variability. Scatter plots show two technical replicates for all quantified proteins, for significantly differentially expressed proteins and also for NSCLC markers used as quality controls in this study (Suppl. Fig. 3B/3C left to right). The average technical correlation for replicate pairs was 0.995 and 0.994 ($p < 0.01$) for ADC and SCC, respectively. The average biological correlation for ADC or SCC tumor within group comparisons were 0.888 and 0.917, respectively.”*

2) The selection of the optimal protocol is not well enough documented. What amount of sample was processed, what is the reproducibility and variability of the data as a function of sample size as the stated goal is needle biopsy level samples.

Response: Thank you for this suggestion. To assure reproducibility, we have now added a comparison of the coefficient of variation (CV) of the proteins quantified in all three protocols across four replicates (see new Fig. 1D, see also manuscript page 5). We observed that the SDS-SP3 protocol has by far the lowest mean CV (0.35) compared to DTR (0.77) and SDC (0.69) protocols. The manuscript has been edited to reflect this change (see manuscript page 5: *“Comparison of the coefficient of variation (CV) of the protein intensities over four replicates per protocol for the proteins identified across all three protocols shows that the SDS-SP3 protocol has by far the lowest mean CV (0.35) compared to DTR (0.77) and SDC (0.69) protocols (Fig. 1D).”*

Processed sample amounts were already described in the Materials and Methods section in the manuscript on page 20.” For the protein extraction protocol comparison, four replicates of

FFPE lung adenocarcinoma tissue samples containing one 10 μm scroll each were used per protocol.” and we have now also included this information in the main text (page 5):” *Four replicates of 1 x 10 μm FFPE slices were processed for each of the three protocols and 1 μg peptide of each sample was injected for LC-MS/MS measurements.*”

We performed the protocol comparison with resected tissue slices to avoid introducing confounding variability as it occurs with diverse sample amounts in clinical biopsies. Based on these results we chose the best-performing protocol, since it can be expected that the protocols with poorer performance for large sample amounts will not perform better for smaller sample amounts.

3) Investigation of fixation time. The number of proteins identified is not really an informative metric. The authors should show a volcano plot to detect potential biases affecting specific protein sets for the conditions compared.

Response: We appreciate your comment and have now included a moderated t-test comparing the protein quantities in the samples that were fixed over-night to those fixed from Friday to Monday and added the results as a volcano plot (see new Suppl. Fig. 1D). Only two proteins were significantly higher expressed (Benjamini-Hochberg-adjusted $p < 0.05$, i.e. 5% FDR) in the over-night samples, CLU and IGHM. The following text was added to the manuscript on page 8:” *A moderated t-test comparing the protein quantities in the samples that were fixed over-night to those fixed from Friday to Monday showed only two proteins significantly upregulated (adjusted p-value < 0.05) in the 24h samples. One was Clusterin (CLU) and the other one immunoglobulin heavy constant mu (IGHM), both variable extracellular proteins that are difficult to quantify (Suppl. Fig. 1D). This shows that there is no relevant bias in protein groups detected after 24h or 72h formalin fixation.*”

4) Generally the paper lacks indications on the amounts of sample required and processed. E.g. dimension of needle biopsy samples actually used and not just “needle biopsy equivalents”.

Response: This information was already included in the original submission, but may have been difficult to find. We would like to point the reviewer’s attention to the following sections: The table in Suppl. Fig. 12B shows the size of a lung FFPE needle biopsy compared to resected tissue and we have described the approximate areas of tumor in the samples on page 12 in the manuscript (“*The equal loading TMT approach is well suited for studies in resected FFPE tissues where the sample amount is not a concern, such as when multiple 10 μm slices of a 5x5 mm^2 tumor section are available.*”) and the protein yields in the table in Suppl. Fig. 12B and in the manuscript on page 17 (“*We could extract, on average, 79 μg of protein material per biopsy. We labeled 20 μg per biopsy sample with TMT and added a 100x booster channel of a reference mix of ADC and SCC samples.*”). Biopsy equivalents are defined in the manuscript on page 12/13: “*To investigate the applicability of the TMT approach for low sample amounts, such as needle biopsy FFPE samples, we utilized 20 μg instead of 200 μg aliquots of the same peptide samples used for the equal loading TMT experiment as “biopsy equivalents”. Protein yield can vary between samples and 20 μg is an amount that should reliably be extracted from a large group of samples.*”

We have now added a description of the materials used for each TMT experiment to the Material and Methods section to make the information about processed sample amounts easier to find (see manuscript page 22: “*For the deep proteome and phosphoproteome analysis of ADC and SCC FFPE tissues, six 10 μm scrolls were combined per sample. From those, 200 μg peptide was used per sample for the equal loading TMT and 20 μg peptide per sample was used as “biopsy equivalents” for the microscaled TMT experiment.*”)

5) Biopsy equivalent --- did they use biopsies or not and for what and with what results? What was actually done for phosphopeptide analysis from needle biopsies? Was the remaining ca. 60 microgram of peptide sample subjected to IMAC enrichment?

Response: We did use actual needle biopsies in addition to the biopsy equivalents and the results are described in the “Applying TMT microscaled proteome/phosphoproteome profiling to clinical FFPE needle biopsies” section of the manuscript, starting on page 17:” *Applying TMT microscaled proteome/phosphoproteome profiling to clinical FFPE needle biopsies*
To test the microscaled approach on clinical FFPE needle biopsies, an additional independent set of eight FFPE needle biopsies consisting of four ADC and four SCC cases was processed with the SDS-SP3 protocol. We could extract, on average, 79 µg of protein material per biopsy. We labeled 20 µg per biopsy sample with TMT and added a 100x booster channel of a reference mix of ADC and SCC samples. In this experiment, 6,800 proteins were quantified from eight FFPE needle biopsies with no missing values, excluding those only found in the boosting channel. [...] For the phosphoproteome, 90% of the TMT-labeled peptide material were used and we could reach a coverage of 5,200 quantified phosphopeptides (Fig. 5B), which is a significant improvement to 1,000 identified phosphopeptides in label-free single-shot LC-MS/MS analyses of FFPE lung needle biopsies (Suppl. Fig. 12B). [...]“
In the revised manuscript, we have now also clarified that 90% of the TMT labelled peptide material were used for phosphoproteome enrichment (manuscript page 17: “*For the phosphoproteome, 90% of the TMT-labeled peptide material were used [...]“*).

6) Generally the number of replicates are too low and too diverse to support confident conclusions. It is recommended to compare replicates of very similar samples e.g. biopsy level samples from the same resected tumor area to determine the technical variability.

Response: We appreciate your concern. We see significant differences between the two lung tumor subtype groups, which indicates that the variability is low enough to detect effects in clinical comparisons. We used known IHC markers and other NSCLC-relevant proteins as quality controls to ensure that the differentially expressed proteins have actual biological relevance.

To address the reproducibility of the microscaled TMT approach we performed as suggested in the general comments an additional TMT experiment with five replicate TMT plexes. We used 20 µg of peptide of consecutive 10 µm FFPE slices from 4 ADC and 4 SCC samples which were randomized into the 8 TMT channels of each plex and then performed deep global proteome and phosphoproteome analysis with the microscaled TMT samples. Each individual TMT plex was analyzed with 30 and 10 high pH RP fractions for the proteome and phosphoproteome, respectively. In total, an additional 20 days of mass spectrometry instrument time were used to evaluate the robustness and reproducibility of the TMT microscaling approach. We can show that mean reporter ion intensities correlate very well between plexes for ADC and SCC proteomes (see new Suppl. Fig. 11b) and we can confidently quantify 8,896 proteins across all plexes on average (see new Suppl. Fig. 11A). On the phosphoproteome level, we can still quantify 4,000 phosphosites across 80% of all tumor samples (see new Suppl. Fig. 11C). Beyond 4 plexes, the quantification rate decreases and we would not recommend adding more TMT plexes, due to the known issue of increasing numbers of missing values with increasing numbers of TMT plexes for very low abundant samples. Nonetheless, the average reporter ion intensities show good correlation for ADC and SCC between all 5 plexes (see new Suppl. Fig. 11D). Please also find a new section on page 16/17 of the manuscript:” *To ascertain the reproducibility of the microscaled TMT approach, we designed five TMT11 plexes out of consecutive 10 µm FFPE slices from eight patients (4 ADC and 4 SCC) [...]. The replicate plexes show an average of 8,896 proteins quantified across all 5 plexes. We quantified >8,100 proteins in all plexes requiring at least 80% valid*

values and >6,500 proteins were quantified among all five plexes with no missing values at all (Suppl. Fig. 11A). [...].

On the phosphoproteome level, we were able to quantify 9,686 phosphosites on average and in these five TMT plexes a reasonable coverage of almost 4,000 quantified phosphosites across 80% of all tumor samples can be achieved (Suppl. Fig. 11C). [...].”

We have also adapted our guidelines on page 21 of the manuscript: “A clear advantage of the microscaled TMT approach presented here is that it provides in-depth coverage on proteome and phosphoproteome level but can tolerate much lower input, hereby enabling comprehensive proteomic characterization of very small clinical specimens such as needle core biopsies (Fig. 6B). Our reproducibility analysis across five microscaled TMT experiments showed a high degree of reproducibility and only minor losses in proteome coverage across plexes. Due to missing value propagation for low input samples across TMT cassettes, we recommend to use the microscaled phosphoproteome approach only for up to four plexes, with up to 64 samples in TMT16-mode, until better methods with improved reporter ion sensitivity are developed. Previous cancer studies with 45 medulloblastoma cases⁵⁴, 27 breast cancer tumors⁵⁵ or 38 prostate cancer samples⁵⁶ and drug response profiling studies with 48 Jak2-mutated neoplasms⁵⁷ show that these cohort sizes can already be useful to molecularly characterize cancer subtypes and help in the discovery of future biomarkers.”

7) The interpretation of the detected differential molecules between ADC and SCC samples is superficial. In addition of highlighting proteins detected as differential that are known as differential in the literature the authors also should describe proteins that are expected to change but were not detected as changed and some effort should be made to assess the differential molecules that are not yet in the literature in this scenario. E.g are these likely genuine differences (e.g. related to known biochemical differences between the samples) or are they artifacts e.g different levels of blood proteins or rather likely contaminants?

Response: To improve the interpretation of ADC and SCC markers, we have now added a new table (Suppl. Table 2) with NSCLC-relevant proteins from literature and indicated by which methods we quantified them. Since many relevant lung cancer proteins are detectable and play a role in both subtypes, only a few lung cancer-associated proteins can be used for differentiating lung cancer subtypes, such as KRT5/6, KRT7 or NAPSAs. All of those proteins were assigned to the correct subtype by our quantitative proteomics analysis. Please also see the scans of IHC staining for KRT5/6 and KRT7 that we have now added as Suppl. Fig. 1A/B for the tissue specificity of those cytokeratins.

We have added, as requested, more information on markers that were described before and compared our proteomics data to results reported in a TCGA RNA expression analysis for the same lung cancer subtypes (Venugopal et al, 2019). Our results confirm these markers on protein level, including MUC5B, TTR, and KNG1 for ADC or APOA1 for SCC.

Our analysis also identified potential new ADC/SCC markers: MUC5AC, CEACAM6, and CSTA and cited their known general associations with lung cancers.

We have now highlighted those additional proteins in the volcano plot in Suppl. Fig. 2A and edited the manuscript on page 6/7 to reflect this change:” *With this dataset, we could confirm several markers on a protein level that had previously been shown on RNA level²³, such as mucin-5B (MUC5B), a glycoprotein that is secreted in the lung and is associated with poor prognosis in ADC²⁴, kininogen 1 (KNG1), and serum transthyretin (TTR), which has been shown to be associated with poor outcome²⁵. In SCC we could confirm apolipoprotein 1 (APOA1) which has previously been described as inversely correlated with risk of lung cancer²⁶. Interestingly, the paper by Venugopal and Yeh et al. (2019) showed fibrinogen alpha (FGA) to be overexpressed in ADC, in our dataset however, it was found with higher expression levels in SCC. Additionally, we were able to quantify several proteins that might be*

of interest as potential future markers for lung ADC, such as mucin-5AC (MUC5AC), another mucin which has been linked to ALK-positive lung adenocarcinoma²⁷, or carcinoembryonic antigen-related cell adhesion molecule 6 (CEACAM6), a glycoprotein that is involved in cell invasion and metastasis and has been shown to have higher expression levels in ADC compared to SCC via IHC²⁸.

Cystatin A (CSTA) has previously been shown to have a higher expression in SCC via IHC²⁹ and might be an interesting potential candidate for SCC.“

To assess artifacts caused by blood contamination, we have compared the intensity distribution of blood proteins in biopsy equivalents (taken from resected tissues with lower blood contamination) and clinical needle biopsies in Suppl. Fig. 13 and the results were very similar and lead us to exclude substantial differences due to blood contaminations. Please see in manuscript on page 30:” Potential blood contamination quality control in needle biopsies *Needle biopsies are taken during bronchoscopy or CT-guided procedures without the possibility to limit blood flow to the target tissue. Therefore, the biopsies often contain more blood than resected tissue FFPE samples. This is not an issue for DNA analyses, since erythrocytes do not contain DNA, but they do contain proteins which in high abundance could influence the identification and quantification of tumor proteins. To investigate the difference in blood content in biopsies and biopsy equivalents, we used a list of 276 proteins identified in dried blood spots by Chambers et al.⁵ as reference. In the biopsy equivalents (derived from resection specimens), 199 proteins out of 276 were identified and 181 out of 276 were identified in the needle biopsies. The intensity distributions of blood proteins in both experiments behave very similarly to the majority of the proteins showing a summed-up intensity between 10^{10} and 10^{11} (Suppl. Fig. 13). Substantial differences between the proteomes of FFPE resected tissues and needle biopsies due to blood contamination can, therefore, be excluded. “*

REVIEWERS' COMMENTS

Reviewer #1 (Remarks to the Author):

The authors have addressed all my concerns.

I also find their responses to the other reviews satisfactory; I have no remaining concerns about this manuscript

Reviewer #2 (Remarks to the Author):

In the revised version of the manuscript authors addressed all major points that were raised by me and other reviewers. Despite the fact that manuscript is not providing any major biological findings, manuscript show mostly technical advancement in the area of needle biopsy analysis by utilizing power of boosting channel with TMT labeling technology. The statistical analysis of label free sample analysis data in comparison to labeled type of sample analysis with and without boosting channel show clear advantage of using such technology for limited amount sample cases. The ability to see deeper into limited sample amount is really great step forward in the areas if minimal sampling for biopsies. I would suggest to publish this manuscript in your journal.

Reviewer #3 (Remarks to the Author):

Summary: The revised paper is substantially improved and the reviewers acknowledges the substantial effort of the authors to respond to the raised issues. The added information with high relevance for clinical studies such as technical and biological variability and a clear indication of the utility of the method for relatively small studies of up to 64 samples will be very useful for the reader. However, several issues remain that should be addressed before publication. These are detailed in the following.

- The statement that the presented method is presently limited to relatively small sample sizes is now explicitly in the discussion/recommendation section. The scope of the method also should be explicitly mentioned in the abstract and introduction.

- Thanks for adding the cv data. It further supports the selection of the SDS-SP3 method as the only usable option among the methods tested.
- The indication of the sample amounts consumed in the different types of analyses is still confusing. What really matters is the volume of tissue processed, the quantity of protein extracted per unit volume or unit weight (e.g xx microgram protein per mg tissue) and the amount of peptide sample injected on column. The authors are requested to provide this information clearly in the text for the different measurements. The indication 1x10 micrometer slice is useless.
- The new data shown in Fig 3 are very informative. It would be useful to assess the effect of eliminating the most variable proteins from the volcano plots.
- Almost 3000 phosphosites were detected in the TMT analyses with booster channel and not in the equal loading TMT measurements. This result is concerning and the explanations given are not convincing. It is suggested to also consider other, technique based possible explanations.
- It is suggested to also mention in the discussion that single shot DIA measurements of small sample amounts have reported protein number and cv results that are substantially higher than the LFQ data reported here, thus pointing to an additional mass spectrometric mode that could be deployed for samples generated by the sample prep workflow.

REVIEWERS' COMMENTS

Reviewer #1 (Remarks to the Author):

The authors have addressed all my concerns. I also find their responses to the other reviews satisfactory; I have no remaining concerns about this manuscript

Reviewer #2 (Remarks to the Author):

In the revised version of the manuscript authors addressed all major points that were raised by me and other reviewers. Despite the fact that manuscript is not providing any major biological findings, manuscript show mostly technical advancement in the area of needle biopsy analysis by utilizing power of boosting channel with TMT labeling technology. The statistical analysis of label free sample analysis data in comparison to labeled type of sample analysis with and without boosting channel show clear advantage of using such technology for limited amount sample cases. The ability to see deeper into limited sample amount is really great step forward in the areas if minimal sampling for biopsies. I would suggest to publish this manuscript in your journal.

Reviewer #3 (Remarks to the Author):

Summary: The revised paper is substantially improved and the reviewers acknowledges the substantial effort of the authors to respond to the raised issues. The added information with high relevance for clinical studies such as technical and biological variability and a clear indication of the utility of the method for relatively small studies of up to 64 samples will be very useful for the reader. However, several issues remain that should be addressed before publication. These are detailed in the following.

- The statement that the presented method is presently limited to relatively small sample sizes is now explicitly in the discussion/recommendation section. The scope of the method also should be explicitly mentioned in the abstract and introduction.

Response: We have added the following sentence to the abstract (page 2):

“Finally, we present general guidelines to which methods are best suited for different applications, highlighting TMT methods for comprehensive (phospho-)proteome profiling for focused clinical studies and label-free methods for large cohorts.”

In addition, we have extended the following sentences in the introduction:

Page 4: *“Nowadays, label-free proteomics methods in which peptides derived from FFPE samples are directly analyzed in a one sample per one LC-MS/MS run manner (“single-shot runs”) can provide quantitative information for between 2,000 and 5,000 proteins^{14,16,17} and are well suited for analysis of large clinical cohorts.”*

Page 5: *“We also provide guidelines on what proteomics/phosphoproteomics methods to use for different sample sets and recommend the use of TMT approaches for comprehensive (phospho-)proteome profiling for focused clinical studies.”*

- Thanks for adding the cv data. It further supports the selection of the SDS-SP3 method as the only usable option among the methods tested.

Response: We thank again the reviewer for this excellent suggestion.

- The indication of the sample amounts consumed in the different types of analyses is still confusing. What really matters is the volume of tissue processed, the quantity of protein extracted per unit volume or unit weight (e.g xx microgram protein per mg tissue) and the amount of peptide sample injected on column. The authors are requested to provide this information clearly in the text for the different measurements. The indication 1x10 micrometer slice is useless.

Response: Sample amount description are described now in more detail for all analysis throughout the manuscript. In detail, on page 5, 22, 23 the description “ca. 150 mm² tumor area on average” was added for resected tissue FFPE samples and “approx. 5 mm² tumor area on average” on page 23 for FFPE needle biopsies.

- The new data shown in Fig 3 are very informative. It would be useful to assess the effect of eliminating the most variable proteins from the volcano plots.

Response: We followed this useful suggestion to exclude non-reproducible proteins before t-testing as illustrated in a new volcano plot. Supplementary Fig. 2A/2B were edited to show proteins that fall outside the 95% prediction intervals in the scatterplots in Supplementary Fig. 3B/3C with an asterisk symbol. The manuscript was edited to reflect this change on page 8 (“*Proteins that fall outside a 95% prediction interval in the scatterplot of technical ADC or SCC replicates are indicated in the volcano plots (Supplementary Fig. 2A/2B) with an asterisk symbol (301 out of 5059 proteins) whereby the variability is usually higher in proteins with lower abundance. We observe that none of the NSCLC-relevant proteins fall outside of these intervals.*”)

- Almost 3000 phosphosites were detected in the TMT analyses with booster channel and not in the equal loading TMT measurements. This result is concerning and the explanations given are not convincing. It is suggested to also consider other, technique based possible explanations.

Response:

The reviewer is of course right and we have added this paragraph to the manuscript on page 15: “*This suggests that the two TMT approaches do not show a bias with respect to the type of phosphopeptides that are enriched and subsequently quantified, but that the differences arise merely from the variability that is known from the stochastic detection of peptides in data-dependent acquisition analysis of low abundant phosphopeptide samples.*”

- It is suggested to also mention in the discussion that single shot DIA measurements of small sample amounts have reported protein number and cv results that are substantially higher than the LFQ data reported here, thus pointing to an additional mass spectrometric mode that could be deployed for samples generated by the sample prep workflow.

Response: DIA measurements seem to be a promising alternative to DDA LFQ approaches and we have included it into the discussion on page 22 (*“Another label-free mass spectrometry approach for the analysis of large retrospective clinical cohorts, that was not explored in this study however, are data-independent acquisition (DIA) methods that were recently applied to quantify approximately 5,000 proteins per sample from several FFPE tissues¹⁴.”*)